# Cycling molecular assemblies for Golgi imaging and disruption

Weiyi Tan [1], Qiuxin Zhang [1], Zhiyu Liu[1], Kangqiang Qiu[2], Divyanshu Mahajan[3], Thomas Gerton[4], Noah Copperman[5], Erica C. Dresselhaus[6], Chaoshuang Xia[7], Cheng Lin[7], William Lau[1], Mikki Lee[1], Isabela Ashton-Rickardt[1], Pengyu Hong [8], Daniela Dinulescu[4], Jer-Tsong Hsieh [9], Avital A. Rodal[6], David M. Loeb[5], Ronny Drapkin [10], Jiajie Diao [2], Lei Lu [3] & Bing Xu [1] ✉

The Golgi apparatus is a central hub for protein trafficking and signaling, yet its rapid imaging and cell-selective disruption remain challenging. Here, we report cycling molecular assemblies (CyMA) for fast Golgi imaging and cell-selective interference. CyMA precursors are acetylated amphiphilic thiopeptides that traverse plasma membrane and are deacetylated by intracellular thioesterases. This exposes thiols that undergo palmitoylation by Golgi-resident palmitoyl acyltransferases utilizing palmitoyl-CoA. The resulting palmitoylated peptides self-assemble into dynamic nanostructures (i.e., CyMA) localized at the Golgi. Their continuous, reversible *S*-acylation enables near-instantaneous Golgi imaging. Replacing fluorophore with a biphenyl motif promotes CyMA accumulation and disrupts functions such as protein modifications, trafficking, and secretion, leading to cell death. This study establishes dynamic supramolecular assembly as an active and selective strategy for Golgi-targeting, pleiotropically interfering with Golgi functions, which may be applicable to targeting other organelles by utilizing alternative enzyme switches to enable kinetic trapping.

Golgi apparatus, as a crucial trafficking and signaling hub in eukaryotic cells, regulates a wide range of cellular functions, including inter-organellar communication[1–3], lipid biosynthesis and trafficking[4,5], cargo processing through glycosylation[6] and lipidation[7], and cell death initiation[8]. It also plays important roles in cancer cell proliferation and metastasis. For instance, several key proteins involved in cancer metastasis, including insulin-like growth factor 1 receptor (IGF-1R)[9], receptor tyrosine-protein kinase erbB-2 (ERBB2)[10], epidermal growth factor receptor (EGFR)[11], and transforming growth factor beta-1 protein (TGFB1)[12] undergo glycosylation in the Golgi, which is essential for their functions within cells. Many cancer drivers[13] and immune checkpoint proteins[14,15] need to be glycosylated or lipidated in the Golgi before performing their functions. Due to its central role, the Golgi is an attractive target for molecular imaging and functional modulation. While some targeting strategies have been reported[16–18], lengthy incubation times and cell selectivity remain significant limitations. Our recent unexpected findings reveal thiophosphopeptides that target the Golgi and selectively kill cancer cells through enzyme-catalyzed, non-equilibrium self-assembly[19]. During a control experiment, we have discovered peptide thioesters that rapidly target the

[1]Department of Chemistry, Brandeis University, Waltham, MA, USA. [2]Department of Cancer Biology, University of Cincinnati College of Medicine, Cincinnati, OH, USA. [3]School of Biological Sciences, Nanyang Technological University, Singapore, Singapore. [4]Department of Pathology, Brigham and Women's Hospital, Harvard Medical School, Boston, MA, USA. [5]Department of Developmental & Molecular Biology and Department of Pediatrics, Albert Einstein College of Medicine, Bronx, NY, USA. [6]Department of Biology, Brandeis University, Waltham, MA, USA. [7]Department of Biochemistry & Cell Biology, Center for Biomedical Mass Spectrometry, Boston University Chobanian & Avedisian School of Medicine, Boston, MA, USA. [8]Department of Computer Science, Brandeis University, Waltham, MA, USA. [9]Department of Urology, Southwestern Medical Center, University of Texas, Dallas, TX, USA. [10]Ovarian Cancer Research Center, University of Pennsylvania, Perelman School of Medicine, Philadelphia, PA, USA. ✉e-mail: bxu@brandeis.edu

Golgi, highlighting the potential of Golgi-targeting assemblies[20]. While implying supramolecular strategies for Golgi targeting, these serendipitous findings also underscore the need to gain a mechanistic understanding and to enhance cell selectivity for desired functions.

Here, we report that cycling molecular assemblies (CyMA) target the Golgi through a futile cycle established by an enzyme switch. An enzyme switch refers to pairs of enzymes that catalyze opposing reactions, such as a kinase/phosphatase pair that regulates phosphorylation and dephosphorylation[21]. In this study, we utilized an enzyme switch composed of counteracting acyltransferases and acyl-protein thioesterases to maintain supramolecular assemblies at the Golgi (Fig. 1a). We refer these supramolecular assemblies as CyMA because they undergo reversible acylation and consume palmitoyl-CoA through a futile cycle. The precursors of CyMA are thioesters of amphiphilic peptide derivatives. They enter cells and are mainly deacylated by thioesterases (TEs), including palmitoyl-protein thioesterase 1 (PPT1), acyl-protein thioesterase 1 (LYPLA1), and acyl-protein thioesterase 2 (LYPLA2))[22] to form thiopeptides. This deacylation enables the subsequent palmitoylation of the thiopeptide by palmitoyl acyltransferases (PATs or zDHHCs)[23,24] (Fig. 1b), leading to kinetic trapping and accumulation of CyMA at the Golgi. According to their N-terminal groups (NTG), we categorize CyMA into two types: CyMA-i for Golgi imaging and CyMA-d for disrupting Golgi functions (Fig. 1c). This reaction cycle allows Golgi imaging at ultralow concentrations of the CyMA-i probes within minutes. Replacing the fluorophore with a biphenyl motif in the CyMA-i generates functional CyMA (CyMA-d), which also undergo the futile cycle of palmitoylation and depalmitoylation. CyMA-d effectively disrupt Golgi functions, including post-translational modifications (PTMs) (such as the palmitoylation or glycosylation of endogenous protein substrates), protein trafficking, and protein secretion. Such Golgi disruption results in inhibition of cell proliferation in cellulo, ex vivo, and in vivo. These assemblies mislocalize critical proteins and disrupt receptor tyrosine kinases (RTKs) signaling. In addition, the designed CyMA-d exhibits cell selectivity, sparing hepatocytes and certain immune cells with high carboxylesterase (CES) expression. In addition to providing a mechanistic understanding of Golgi-targeting by thiophosphopeptides[19] and peptide thioesters[20], this work introduces a previously unexplored strategy that leverages enzyme switches and reversible lipidations of proteins to kinetically trap supramolecular assemblies at specific organelles, which may have broad applications for organelle-specific imaging, immune modulation, metabolic regulation, and beyond.

## Results

### Molecular design

Our previously reported Golgi targeting thiophosphopeptides[19] and peptide thioesters[20] share three common features: (i) An environmental-sensitive fluorophore (nitrobenzoxadiazole (NBD)) that becomes highly fluorescent in assembled states[25]. (ii) A dipeptide (D-diphenylalanine (D-Phe-D-Phe)[26]) that resists proteases and favors self-assembly[27]. (iii) An enzyme-responsive trigger (thiophosphate or thioester) that enables enzyme-instructed self-assembly[28] upon catalysis of the respective enzyme (alkaline phosphatases (ALP) or thioesterases). While the thiophosphopeptide exhibits excellent cell selectivity based on the expression level of ALP[19], it undergoes auto-dephosphorylation in acidic aqueous solution, limiting its broad applicability. Conversely, peptide thioesters, despite their superior chemical stability and ability to image and target the Golgi in various cell lines[20], lack cell selectivity. Therefore, it is necessary to develop a stable and cell-selective molecular platform for Golgi targeting.

To improve stability and selectivity, we designed the Golgi-targeting molecular platform, CyMA, by incorporating an ester linkage in the peptide thioesters, according to two main considerations. First, nature has evolved peptides that contain both amide bonds and ester bonds, as exemplified by a class of natural products known as depsipeptides[29,30]. The development of depsipeptide synthesis provided a solid chemistry foundation for making CyMA that contains an ester bond in the peptide framework. Second, the presence of an ester bond allows CyMA to be substrates of both amidases and esterases. The high expression of esterases in liver cells would be an effective mechanism to hydrolyze the carboxylester bond, thus minimizing the impact of CyMA on hepatocytes.

To construct the CyMA scaffold, we capped the N-terminus of a short peptide with an N-terminal group (NTG) to generate a self-assembling motif. We then introduced an ester bond into the peptide backbone by conjugating 2-mercaptoethanol directly at its C-terminus, followed by thioesterification of the thiol group to generate the CyMA precursors (Fig. 1c). For the development of CyMA-i, **1a** and its analogs, the fluorophore (Supplementary Fig. 1) acts as their NTG. We changed the acetyl thioester of **1a** into free thiol (**2a**), acetyl ester (**2b**), hydroxyl (**2c**) and sulfonic acid (**2 d**) for illustrating the significance of the thioester bond in the precursors for targeting the Golgi. We further synthesized the non-fluorescent CyMA-d precursor, **3a**, by changing the NBD of **1a** with a more hydrophobic NTG, biphenyl (Fig. 1c). We expect CyMA-d to disrupt Golgi functions. Following the rationales for creating **2a**, **2b** and **2c**, we transformed the acetyl thioester of **3a** into free thiol, acetyl ester, and hydroxyl to produce **4a**, **4b**, and **4c**, respectively, as controls of **3a**. Furthermore, to evaluate the role of the ester linkage in regulating CyMA dynamics, we replaced the ester bond in **1a** and **3a** with a more stable amide bond, generating peptide thioester **5** and **6**, respectively. These compounds serve as key controls to underscore the selectivity conferred by the ester bond in the CyMA precursors.

### Synthesis of CyMA precursors

The thiol group on 2-mercaptoethanol is selectively attached to the resin because of its higher nucleophilicity than the hydroxyl group. The ester bond is formed between the hydroxyl group from resin-bound 2-mercaptoethanol and the carboxylic group from the Fmoc-protected amino acid by Steglich esterification. After the standard solid-phase peptide synthesis (SPPS), the thiodepsipeptide, being cleaved from the resin, is directly reacted with acetyl chloride to generate a thioester bond with a high yield (Supplementary Fig. 2). The crude CyMA precursors are further purified by reverse-phase HPLC to give the final CyMA precursors. A similar process is used to synthesize the control molecules (Supplementary Fig. 3).

### Golgi targeting by CyMA

We first examined the intracellular location of **1a**. Bright green fluorescence from **1a** colocalized with GALNT2-RFP, a Golgi-resident protein, with a Pearson correlation coefficient (PCC) of $0.89 \pm 0.017$ (Fig. 2a). The fluorescence hardly overlapped with lysosomes (Supplementary Fig. 4a), mitochondria (Supplementary Fig. 4b), or ER (Supplementary Fig. 4c). Golgi morphology became visible within one minute of treatment with 500 nM of **1a** (Fig. 2b and Supplementary Movie 1), and **1a**'s Golgi fluorescence was detectable at even lower concentrations (e.g., 200 nM or 50 nM) (Supplementary Fig. 4d and Supplementary Movie 1). Besides mammalian cells, **1a** also targets the Golgi in *Drosophila* cells (Supplementary Fig. 4e). In contrast, MGCTLSA-NBD (Supplementary Fig. 4f) (10 μM), a conjugate of NBD with peptide MGCTLSA that governs $G_{\alpha o}$ localization at the Golgi[31], displayed minimal Golgi targeting with only a few puncta in the peri-nucleus region (Supplementary Fig. 4f). These results confirm the specificity and efficacy of CyMA as Golgi imaging probes.

While **1a** colocalizes with GalT-RFP at 18 °C (Supplementary Fig. 4g), a temperature that specifically arrests the endosome-to-Golgi transport[32], no **1a** enters the cell at 0 °C. This result suggests that **1a** entry into the cell is energy dependent but independent of the canonical Vps35-retromer pathway, as evidenced by higher (rather than lower) cellular uptake in VPS35-mutant *Drosophila* hemocytes

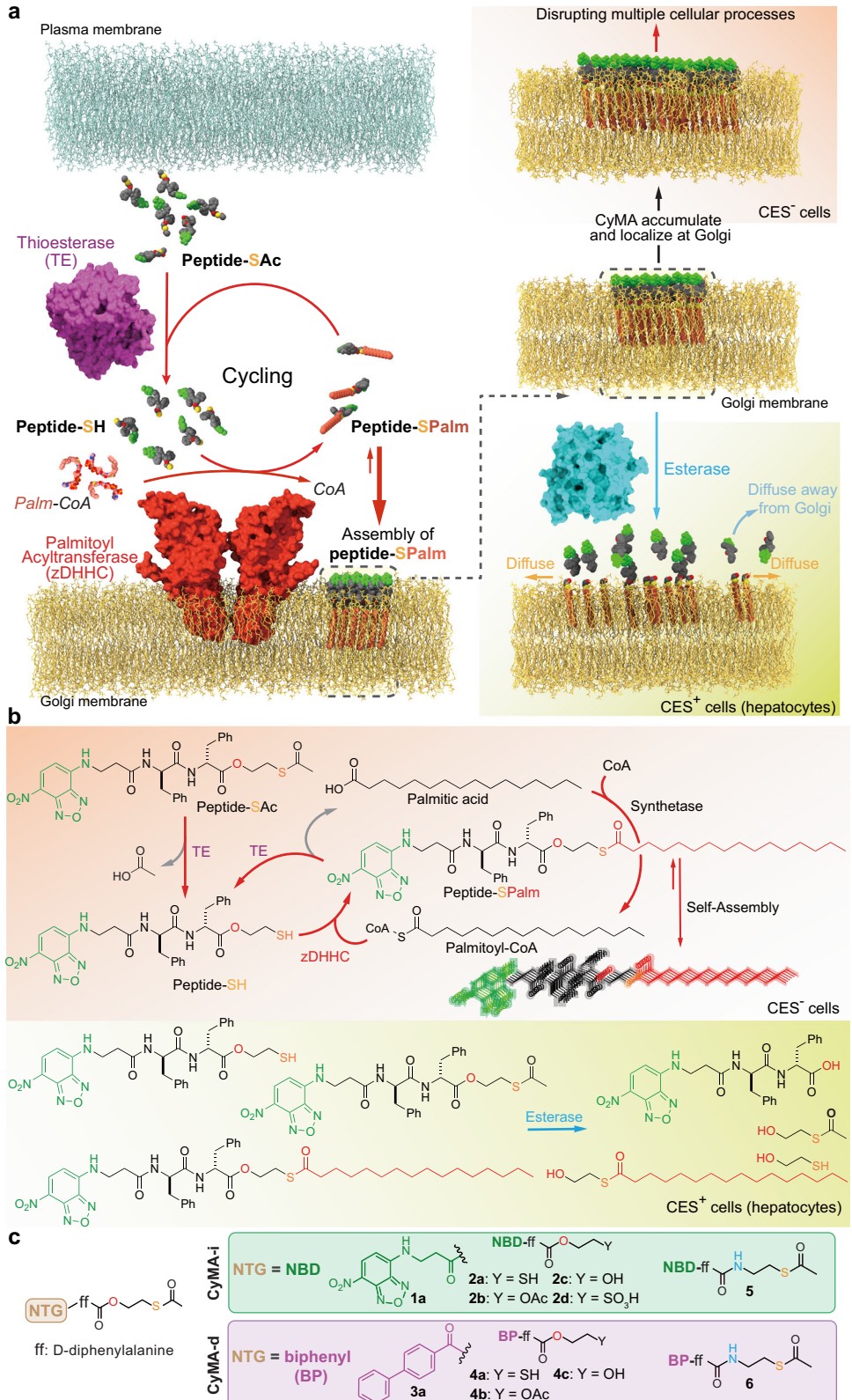

**Fig. 1 | Selective accumulation of CyMA at the Golgi through an enzyme switch to establish futile cycles. a** Schematic illustration of the futile cycle initiated by enzymatic deacylation of CyMA precursors, followed by selective accumulation at the Golgi through an enzyme switch involving thioesterases and palmitoyl acyltransferases. The consequences of CyMA accumulation at the Golgi in CES$^{-/+}$ cells are illustrated. **b** Molecular structures and involved reaction pathways of CyMA precursors and metabolites within the futile cycle, along with exit from the cycle via hydrolysis of CyMA by esterases. **c** Molecular structures of CyMA (CyMA-i and CyMA-d) precursors, relevant analogs and control compounds.

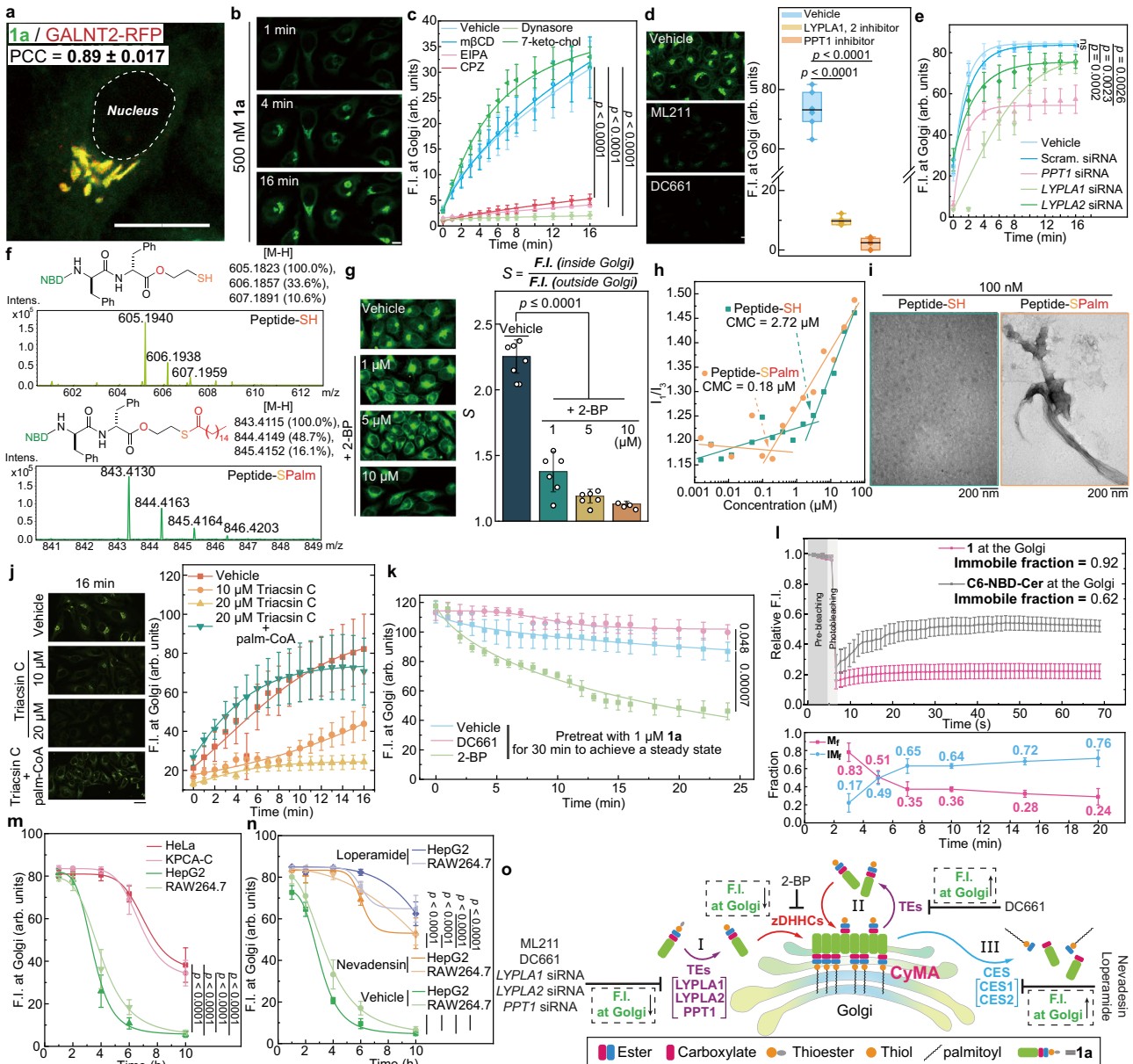

**Fig. 2 | Golgi targeting features of CyMA precursors and the role of enzymes in futile cycle creation and disruption. a** CLSM and colocalization of GALNT2-RFP HeLa cells treated with **1a** (1 µM, 10 min) (*n* = 6). **b** CLSM of HeLa treated with **1a** (500 nM) at indicated times. **c** Golgi fluorescence intensity in HeLa pretreated with endocytosis inhibitors and treated with **1a** (500 nM) (*n* = 6). **d** CLSM and Golgi fluorescence in HeLa cells pretreated with vehicle, ML211, or DC661, then treated with **1a** (1 µM, 1 h) (*n* = 6). **e** Golgi fluorescence in HeLa transfected with indicated TE siRNA and treated with **1a** (5 µM) (*n* = 5). **f** HRMS of **1a** and its palmitoylated form. **g** CLSM and "Golgi-specificity" analysis of HeLa cells pretreated with vehicle or DHHC inhibitor (2-BP) for 30 min and treated with **1a** (2 µM, 20 min). **h** CMC determination of peptide-SH form and peptide-SPalm form of **1a**. **i** TEM of peptide-SH form and peptide-SPalm form of **1a**. **j** CLSM and the Golgi fluorescence intensity of HeLa

pretreated with vehicle, Triacsin C (10 or 20 µM), Triacsin C (20 µM) + palmitoyl-CoA (50 µM) for 30 min, and incubated with **1a** (500 nM) (*n* = 10). **k** Golgi fluorescence in HeLa pretreated with **1a** (1 µM, 30 min), switched to media with DC661 or 2-BP (*n* = 6). **l** FRAP and mobile/immobile fraction analysis of Golgi after **1a** (2 µM, 30 min) or C6-NBD-Ceramide (10 µM, 1 h) treatment (*n* = 10). **m** Golgi fluorescence in CES-negative (HeLa, KPCA-C) and CES-positive (HepG2, RAW264.7) cells treated with **1a** (2 µM) (*n* = 6). **n** Golgi fluorescence intensity in CES-positive cells (HepG2, RAW264.7) pretreated with vehicle, Nevadensin, or Loperamide, and treated with **1a** (1 µM) (*n* = 6). **o** Schematic of CyMA assembly/disassembly dynamics. Scale bar = 20 µm. Data are mean ± s.d. (*n*-values as indicated in the panel). Statistical significance was determined by a two-tailed Student's *t* test. Reproducibility and statistical details are provided in the Methods. Source data are provided as a Source Data file.

(Supplementary Fig. 4h). This increased accumulation may result from the dual role of the Vps35 retromer complex, which helps traffic cargo from the endosome to both the Golgi and the plasma membrane for efflux. Since **1a** utilizes a Vps35-independent mechanism to reach the Golgi, the blockage of the Vps35-dependent efflux pathway to the plasma membrane leads to a higher intracellular concentration of the probe, resulting in its increased accumulation at the Golgi. The endocytosis inhibitors (chlorpromazine (CPZ), 5-(N-ethyl-N-

isopropyl)-amiloride (EIPA) or dynasore) impaired the fluorescence increase at the Golgi, while a caveolin-mediated endocytosis inhibitor, methyl-beta-cyclodextrin (mβCD), and a CLIC/GEEC endocytosis pathway inhibitor, 7-keto cholesterol (7-keto-chol), hardly inhibit the endocytosis of **1a** to accumulate at the Golgi. These results indicate that **1a** enters cells via multiple pathways, including clathrin- and dynamin-mediated endocytosis[33,34], macropinocytosis (Fig. 2c)[35], and endosome-independent pathways[36]. The amphiphilic and

hydrophobic nature of **1a** likely allows it to traverse membranes[37]. As a small molecule, it can translocate from within endosomal compartments to the cytosol, even if initially internalized via clathrin-mediated endocytosis. Importantly, since there is no known endocytic process that specifically delivers material from the extracellular space to the Golgi within a half-time of less than 5 minutes, it is likely that the CyMA precursors, being amphiphilic molecules, bypass canonical endosomal trafficking and traverse the plasma membrane to rapidly access the Golgi. Although the precursor molecule (**1a**) has a limited hydrophilic part, the polar peptide backbone renders it amphiphilic and facilitates cell entry.

### Deacylation by TEs activates CyMA precursors

To verify that TEs hydrolyze the thioester bond in **1a**, we treated HeLa cells with specific inhibitors for PPT1, LYPLA1, and LYPLA2. Pretreatment with ML211 (LYPLA1/2 inhibitor)[38] significantly reduced Golgi fluorescence, and DC661 (PPT1 inhibitor)[39] decreased **1a** accumulation (Fig. 2d). Both inhibitors had a minimal impact on the Golgi accumulation of C6-NBD-Ceramide, and the Golgi structure remains intact following inhibitor pretreatment (Supplementary Fig. 4i). In addition, siRNA knockdown of *PPT1* led to a comparable rate of Golgi accumulation but resulted in a lower plateau of Golgi fluorescence intensity (Fig. 2e and Supplementary Fig. 4j). Furthermore, the knockdown of *LYPLA1* reduced the rate of fluorescence build-up at the Golgi, whereas *LYPLA2* knockdown had a lesser effect (Fig. 2e and Supplementary Fig. 4j), suggesting that LYPLA1 might play a more significant role than LYPLA2 in HeLa cells. The distinct patterns of Golgi accumulation observed following *PPT1* and *LYPLA1* knockdown may be attributed to differences in their enzymatic activities and subcellular localizations.

### CyMA are palmitoylated by zDHHCs

Since the thiophosphate (NBD-ff-pS)[19], peptide thioester (NBD-ff-SAc)[20], and **1a** all require enzymatically forming thiols for rapid Golgi targeting, the resulting thiopeptide (NBD-ff-SH) or thiopeptide derivative (**2a**, Fig. 1c) likely would be the active species responsible for Golgi targeting. It is possible that the thiol group in **2a** undergoes palmitoylation, leading to its accumulation at the Golgi. This assumption is supported by several studies demonstrating the palmitoylation of cysteine-containing peptides[17,18] and recent reports implying the palmitoylation of thiol-containing Golgi-trackers[40,41]. To obtain direct evidence of thiopeptide derivative palmitoylation, we analyzed the lysates of HeLa cells treated with **1a**. Liquid chromatography coupled to high-resolution mass spectrometry (LC-HRMS) confirmed CyMA lipidation at the Golgi by identifying peaks for both the deacylated product (Peptide-SH) and the palmitoylated CyMA (Peptide-SPalm) of **1a** (Fig. 2f and Supplementary Fig. 4k). Notably, the palmitoylated CyMA exhibited a retention time comparable to that of the chemically synthesized palm-**1a** (Supplementary Fig. 4l), further confirming that CyMA undergo palmitoylation. Under these mild conditions (1 μM for 30 min), we did not detect other potential products, such as those from alternative S-lipidation or disulfide formation, suggesting that the rapid, enzyme-driven *S*-palmitoylation is the dominant pathway for the free thiol intermediate.

Inhibiting zDHHCs by 2-bromopalmitate (2-BP)[42] reduced CyMA accumulation at the Golgi in a dose-dependent manner (Fig. 2g) (with no signs of reaction happening between CyMA and 2-BP (Supplementary Fig. 4 m)), supporting the role of palmitoylation (i.e., switching from Peptide-SH to Peptide-SPalm) in Golgi localization. 2-BP showed similar inhibition on CyMA with free thiol groups (**2a**) (Supplementary Fig. 4n), but no obvious inhibition on C6-NBD-ceramide (Supplementary Fig. 4i), confirming that palmitoylation is essential for Golgi accumulation of CyMA.

A critical consequence of this intracellular modification is the triggering of supramolecular self-assembly. The transformation from Peptide-SH to the amphiphilic Peptide-SPalm induced a dramatic, over

15-fold decrease in the critical micelle concentration (CMC) from 2.72 μM to 0.18 μM (Fig. 2h). This propensity for self-assembly was confirmed by transmission electron microscopy (TEM), which revealed that Peptide-SPalm (palm-**1a**) readily forms distinct nanostructures, whereas the precursor Peptide-SH (**2a**) remains soluble with no visible aggregates (Fig. 2i).

CyMA hijack the cell's endogenous biosynthetic machinery and building blocks. While siRNA knockdown of several zDHHC enzymes suggested a degree of enzyme redundancy (Supplementary Fig. 4o), we found the process was critically dependent on the intracellular supply of palmitoyl-CoA. Inhibiting the synthesis of long-chain fatty acyl-CoA by triacsin C[43] significantly impaired Golgi localization of **1a** (Fig. 2j). This effect was fully rescued by the exogenous addition of palmitoyl-CoA, confirming it as the specific acyl donor (Fig. 2j). These results demonstrate a strategy to the design of self-assembling amphiphilic molecules, wherein in situ activation is achieved by harnessing the cell's endogenous enzymatic switches (TEs and zDHHCs) alongside endogenous metabolic substrates (palmitoyl-CoA), thereby facilitating organelle-specific localization and self-assembly.

### Reversible S-acylation of CyMA

To verify the palmitoylation-depalmitoylation cycle of CyMA at the Golgi, we treated cells with DC661 and 2-BP, the inhibitors for TEs and zDHHCs, respectively, after first accumulating **1a** at the Golgi. Removing **1a** from the media caused fading of Golgi fluorescence, reflecting the balance between Peptide-SPalm and Peptide-SH (Fig. 2k), in other words, a steady state. Inhibiting TEs increased fluorescence retention, while inhibiting zDHHCs reduced it. The varying impacts of TE inhibition on fluorescence intensity at different time points may appear contradictory (Fig. 2d, k). However, this demonstrates the futile cycle at the Golgi. Early TE inhibition halts the deacetylation of **1a** to Peptide-SH, blocking palmitoylation and Golgi accumulation, thus reducing fluorescence. As deacetylation proceeds and the palmitoylation-depalmitoylation cycle begins, TE inhibition maintains fluorescence by increasing the Peptide-SPalm/Peptide-SH ratio. Longer incubation (Supplementary Fig. 5a, b) and repeating the experiments with the CyMA precursor (**5**) having an amide bond linker yielded the same outcomes (Supplementary Fig. 5c), ruling out the possibility that the fluorescence decrease was due to the action of esterases. The palmitoylation-independent Golgi probe, C6-NBD-ceramide, exhibited a similar rate of fluorescence fading under either inhibitor (Supplementary Fig. 5d), further confirming the unique feature of CyMA.

We synthesized palmitoylated CyMA by substituting the acetyl group of **1a** or **5** with a palmitoyl group (Supplementary Fig. 5e), and these synthetic peptide-SPalm exhibit limited cellular entry and tend to form aggregates extracellularly because of their high hydrophobicity. These findings confirm the continuous palmitoylation-depalmitoylation cycles of CyMA at the Golgi and that in situ palmitoylation is crucial for the Golgi localization of CyMA.

### CyMA is non-diffusive

To assess the diffusion of CyMA, we conducted fluorescence recovery after photobleaching (FRAP) on CyMA localized at the Golgi. A concentration of 2 μM of **1a** was selected, as it does not induce Golgi fragmentation under these conditions. The results showed that the photobleached region remained dark after 60 seconds. The mobile fraction ($M_f$) was 0.08, and the immobile fraction ($IM_f$) was 0.92, indicating CyMA are solid-like and hardly diffuse from the Golgi. The time-dependent increase of $IM_f$ correlated with CyMA accumulation. Control FRAP with C6-NBD-ceramide[16] showed faster fluorescence recovery and yielded a lower $IM_f$ of 0.62 (Fig. 2l), suggesting that CyMA, rather than Golgi compartmental stability[44], account for the high $IM_f$. The high immobile fraction, indicative of a solid-like state, can be reconciled with the dynamic, reversible *S*-palmitoylation cycle by

distinguishing the behavior of individual molecules from the bulk assembly. The dynamic cycling represents the constant turnover of individual molecules at the interface of the assembly, while the high immobile fraction reflects the stability of the large, macroscopic core that is kinetically trapped at the Golgi. This model also explains the assembly's net disassembly upon precursor removal (Fig. 2k), which disrupts the system's steady state.

## Cell selectivity of CyMA

To establish the cell selectivity conferred by introducing the ester bond, we examined how esterases affect CyMA accumulation at the Golgi in cell lines with varying CES expression levels. In HepG2 and RAW264.7 cells (high CES expression), Golgi fluorescence significantly decreased after 4 hours of **1a** treatment, while it remained unchanged in HeLa and KPCA-C cells (low CES expression) (Fig. 2m and Supplementary Fig. 6a). CES1 and CES2 inhibitors preserved Golgi fluorescence in HepG2 and RAW264.7 cells (Fig. 2n), while having little effect in HeLa cells (Supplementary Fig. 6b). Treatment with compound **5**, the amide linker analog of **1a**, maintained Golgi fluorescence in HepG2 and RAW264.7 cells after 4 h, regardless of CES1 or CES2 inhibition (Supplementary Fig. 6c). These results confirm that esterases regulate the dynamics and accumulation of CyMA at the Golgi, supporting the processes illustrated in Fig. 1a, b.

## Mechanism of Golgi targeting by CyMA

The above results of **1a** and its analogs, collectively, indicate that CyMA actively target the Golgi via three distinct steps (Fig. 2o): (I) TEs hydrolyze the thioester bond in CyMA precursors, producing thiol assemblies. (II) Thiol $S$-palmitoylation by zDHHCs at the Golgi[7] promotes the formation of non-diffusive assemblies, while TEs reverse this process via depalmitoylation, leading to continuous cycling, with strong green fluorescence (from the NBD-containing assemblies) observed at the Golgi. (III) Esterases hydrolyze the ester bond, generating hydrophilic molecules that disassemble and diffuse from the Golgi. Inhibiting TEs reduces initial fluorescence, while inhibiting esterase increases it. During the reversible $S$-palmitoylation cycles, inhibiting zDHHCs reduces fluorescence, while inhibiting TE enhances it. Since thiophosphopeptides[19] or peptide thioesters[20] form thiopeptides after enzymatic activation, and these thiopeptides enter the futile cycles, the mechanism shown in Fig. 2o is applicable to them, although their activating enzymes differ.

## CyMA as a Golgi probe for various cells

To examine their applicability, we tested CyMA for Golgi imaging on a variety of cell lines. Upon treatment with 500 nM of **1a**, the Golgi becomes clearly labeled within 4 min, with fluorescence intensity gradually increasing and reaching saturation within 30 min (Fig. 3a). The rate of Golgi-specific fluorescence accumulation varies among cell lines, likely due to differences in enzyme levels and cellular environment (Fig. 3b). These results confirm that CyMA-i precursor **1a** functions as a general Golgi probe across various cell lines.

To investigate the key structural requirements for Golgi targeting, we replaced the acetyl thioester of **1a** with a free thiol (**2a**), acetyl ester (**2b**), hydroxyl (**2c**), and sulfonic acid (**2 d**). At high concentrations (10 μM), **1a** rapidly accumulated at the Golgi and saturated throughout the entire HeLa cell in 8 min. In contrast, **2a** was internalized less efficiently, though accumulated at Golgi (Fig. 3c). **2b** showed minimal internalization and primarily localized to the cytosol, and neither **2c** nor **2 d** were taken up by cells or accumulated in the Golgi (Fig. 3c), underscoring the critical role of the thiol group in enabling enzymatic $S$-palmitoylation and efficient Golgi targeting. These results also indicate that the NBD-diphenylalanine motif alone does not drive Golgi localization, ruling out non-specific or specific protein binding as the primary mechanism.

Importantly, **2a** produced significantly less fluorescence at the Golgi than **1a** under identical conditions. Even after 30 minutes of treatment with 500 nM **2a**, only weak perinuclear fluorescence was observed (Fig. 3d). Although our LC-HRMS analysis confirmed that **2a** is a substrate for palmitoylation (Supplementary Fig. 4k), its lower efficacy suggests that when administered exogenously, free thiols, prior to reaching the Golgi, may engage in additional reactions[40] beyond those that lead to Golgi localization, such as the formation of disulfide bonds with proteins or being oxidized. This suggests that in-situ thiol formation and reversible $S$-palmitoylation—as in the case of **1a**—are key to achieving high targeting efficiency.

Furthermore, we replaced the NBD fluorophore in **1a** with other fluorophores, such as dansyl (DAN) or 4-(N,N-dimethylsulfamoyl)-2,1,3-benzoxadiazole (DBD) to generate **DAN-CyMA-i** and **DBD-CyMA-i**, respectively. These derivatives enabled imaging under different excitation wavelengths (Fig. 3e), demonstrating the modularity and versatility of the CyMA-i scaffold for Golgi imaging.

## CyMA as a Golgi probe in vivo

To evaluate the Golgi-targeting performance of CyMA in vivo, we tested **1a** in *Drosophila* larvae. A 10 min treatment of **1a** resulted in high accumulation in larval muscle tissues, with lower uptake observed in the brain and salivary glands (Fig. 3f). This suggests potential tissue selectivity, likely influenced by enzyme distribution and activity differences across organs. Importantly, **1a** exhibits good colocalization with mannosidase II (ManII) in larvae muscle cells (Fig. 3g), confirming its ability to label the Golgi in vivo.

## CyMA-d disrupt Golgi and affect its interacting organelles

In contrast to CyMA-i, which was designed as a minimally perturbing Golgi imaging tool (Supplementary Fig. 7a), CyMA-d incorporates a more hydrophobic NTG, biphenyl, known to enhance membrane affinity[45,46]. This structural modification enables CyMA-d to modulate Golgi functions. To investigate the morphological and functional consequences of such modulation, we used compound **3a**, a representative CyMA-d precursor. To confirm that this non-fluorescent compound localizes to the Golgi, we provide two lines of evidence. First, a structurally similar fluorescent analogue (**3a'**), in which one benzene ring of the phenylalanine residue was replaced with an NBD fluorophore, showed clear and rapid accumulation at the Golgi (Supplementary Fig. 7b). Second, our LC-HRMS analysis of cell lysates confirmed that **3a** is palmitoylated (Supplementary Fig. 7c), providing strong biochemical evidence for its localization at the Golgi, where the required zDHHC enzymes reside[7]. We first evaluated the Golgi integrity following treatment of **3a**. KDELR2, an ERGIC/cis-Golgi protein, became dispersed, and markers of the medial- and trans-Golgi also exhibited scattered localization, but none of these Golgi proteins fully redistributed into the ER (Fig. 4a). This behavior differs from that caused by brefeldin A (BFA), a classical ER-to-Golgi trafficking inhibitor that typically causes complete Golgi collapse into the ER[47]. This highlights a unique mechanism of trafficking disruption by CyMA-d that leads to the loss of Golgi integrity without full ER redistribution. The disruption of the Golgi by CyMA-d also perturbs other organelles that interact with the Golgi. Upon treatment with **3a**, the ER becomes increasingly entangled, suggesting ER stress or structural remodeling (Fig. 4b). Mitochondria adopt rounded morphologies consistent with mitochondrial fission (Fig. 4c). These results indicate that CyMA-d results in broader organelle disruption, which is likely due to organelle membrane disruption[48].

## CyMA-d disrupt anterograde trafficking

We determined the anterograde trafficking with the Retention Using Selective Hooks (RUSH) system[49]. We monitored the ER-to-Golgi trafficking of ManII, TNF-α, and E-cadherin, as well as Golgi-to-plasma

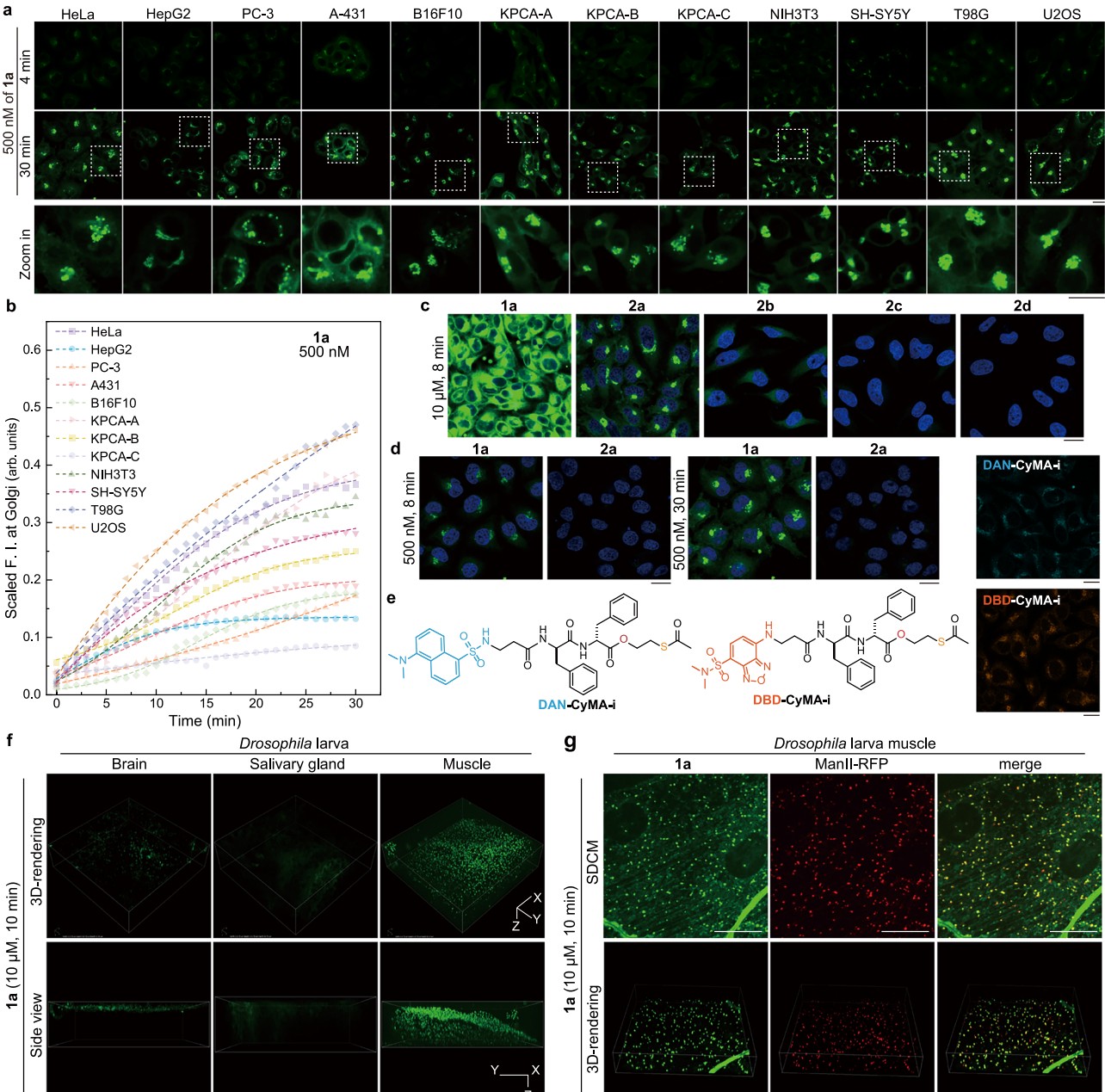

**Fig. 3 | CyMA-i as a universal Golgi probe in cellulo and in vivo. a** CLSM of various cell lines treated with **1a** (500 nM, 4, 30 min). **b** Golgi fluorescence intensity in various cell lines treated with **1a** (500 nM, 0–30 min) ($n \geq 6$). **c** CLSM of HeLa cells treated with **1a**, **2a**, **2b**, **2c**, **2 d** (10 μM, 8 min). **d** CLSM of HeLa cells treated with **1a**, **2a** (500 nM, 8, 30 min). **e** Molecular structures of DAN- and DBD-conjugated CyMA-i, with corresponding CLSM images of HeLa cells treated with DAN-CyMA-i (500 nM, 30 min) or DBD-CyMA-i (5 μM, 15 min). **f** 3D-rendering and side view of brain, salivary gland, and muscle of *Drosophila* larva treated with **1a** (10 μM, 10 min). **g** Colocalization study of **1a** with Golgi marker (C57-Gal4-driven UAS-ManII-RFP) in muscle cells of *Drosophila* larva treated with **1a** (10 μM, 10 min). Scale bar = 20 μm. Data are mean ± s.d. (*n*-values as indicated in the panel). Statistical significance was determined by a two-tailed Student's *t* test. Reproducibility and statistical details are provided in the Methods. Source data are provided as a Source Data file.

membrane (PM) trafficking of TNF-α and E-cadherin (Fig. 4d, e). In control cells, all three cargo proteins exit the ER and localize to the Golgi within 10 min of biotin addition. However, in cells pretreated with **3a**, the proteins remained in the ER, and no forward trafficking is observed (Fig. 4d), indicating that CyMA-d effectively block ER-to-Golgi trafficking.

TNF-α and E-Cadherin, posttranslationally modified in the Golgi before being transported to the PM[49], were used to study Golgi-to-PM trafficking. We kept cells at 20 °C with biotin to accumulate these proteins in the Golgi, followed by treatment with **3a**. Upon shifting to 37 °C, we observed a decrease in Golgi fluorescence (Fig. 4e) for TNF-α

and E-Cadherin in untreated cells, while no significant fluorescence decrease observed in the Golgi of CyMA-treated cells, indicating that Golgi-to-PM trafficking was also impaired.

## CyMA-d disrupt retrograde trafficking

To examine retrograde trafficking from the plasma membrane to the Golgi, we tracked the internalization of fluorophore-labeled cholera toxin subunit B (CTxB), a classical retrograde tracer[50] (Fig. 4f). In control cells, CTxB localized to the Golgi and colocalized with Giantin[51]. In **3a**-treated cells, CTxB showed reduced colocalization with Giantin, suggesting compromised retrograde transport from the PM to

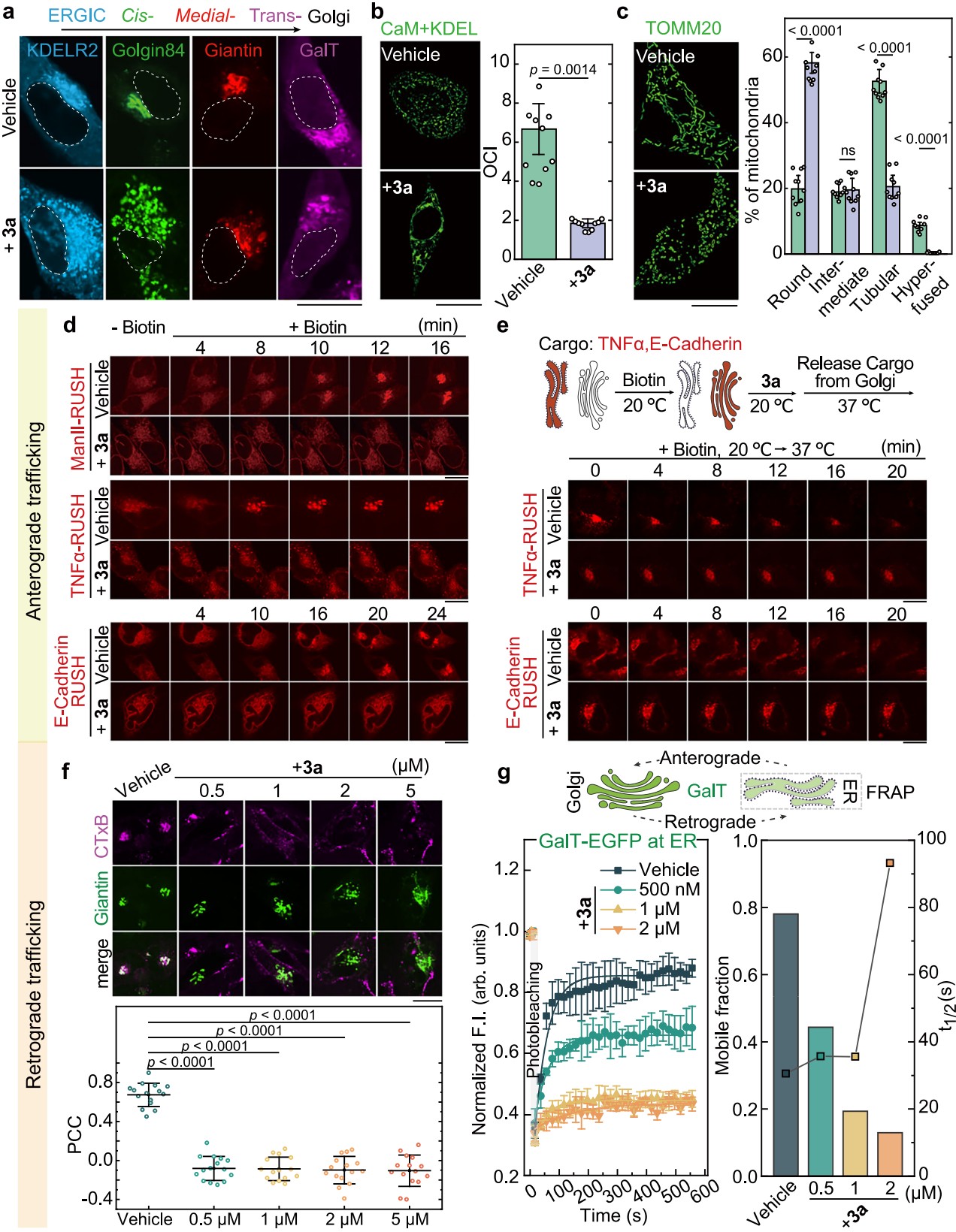

the Golgi (Fig. 4f). We also evaluated Golgi-to-ER retrograde trafficking using FRAP of the ER pool of B4GALT1 (GalT), which shuttles between the ER and Golgi[52] (Fig. 4g and Supplementary Fig. 8). Treatment with **3a** led to a concentration-dependent inhibition of GalT mobility, evidenced by a reduced mobile fraction and increased $t_{1/2}$ (Fig. 4g), demonstrating impaired Golgi-to-ER trafficking.

Collectively, the above results indicate that CyMA-d molecules accumulate at the Golgi and eventually reaches a steady state, leading to disruption of Golgi-associated trafficking, as well as interruption of ER homeostasis and mitochondrial fission. Unlike perturbing Golgi-related trafficking through the decoration of target organelles with specific golgins[53], CyMA-d interfere with both

**Fig. 4 | CyMA-d at Golgi disrupt intracellular trafficking. a** CLSM images of KDELR2, Golgin84, Giantin and GalT in HeLa cells treated with or without **3a** (2 μM, 6 h). **b** Structured illumination microscopy (SIM) images of the ER in HeLa cells treated with or without **3a** (2 μM, 24 h). **c** SIM images of mitochondria in HeLa cells treated with or without **3a** (2 μM, 24 h). **d** Time-lapse CLSM of ManII-mCherry-RUSH, TNFα-mCherry-RUSH, and E-Cadherin-mCherry-RUSH in HeLa cells treated with or without **3a** (2 μM, 6 h), followed by biotin (40 μM) treatment. **e** Schematic illustration of Golgi-to-PM trafficking determination using the RUSH system, and time-lapse CLSM of TNFα-mCherry-RUSH and E-Cadherin-mCherry-RUSH in HeLa

cells treated with or without **3a** (2 μM, 6 h) at 20 °C, followed by monitoring at 37 °C. **f** CLSM of CTxB-AF647 and Giantin-mNeonGreen in HeLa cells pretreated with or without **3a** at various concentrations for 6 h, then treated with CTxB-AF647 (1 μg/mL, 1 h), and the colocalization analysis ($n = 15$). **g** FRAP analysis of the ER pool of GalT-EGFP with or without the treatment of **3a** at different concentrations for 6 hours, showing the quantified mobile fraction and $t_{1/2}$. Scale bar = 20 μm. Data are mean ± s.d. (*n*-values as indicated in the panel). Statistical significance was determined by a two-tailed Student's *t* test. Reproducibility and statistical details are provided in the Methods. Source data are provided as a Source Data file.

anterograde and retrograde transports to and from the Golgi, a consequence of the dysfunctional trafficking resulted from its dynamic accumulation and supramolecular assembly at the Golgi.

## Golgi disruption affects posttranslational modification (PTM)

As a central organelle for PTMs, the Golgi plays a critical role in ensuring proper protein function. Disruption of the Golgi by CyMA-d is therefore expected to interfere with key PTMs, including protein lipidation and glycosylation. We investigated these effects using metabolic labeling (Fig. 5a)[54–56] and analyzed the outcomes through CLSM and LC-MS/MS. We observed a concentration-dependent reduction in palmitoylated proteins, indicating that CyMA-d impair lipidation (Fig. 5b). While this metabolic labeling strategy is a widely established method[57–59], we acknowledge that the probe can also be incorporated into cellular lipids. To provide more specific and robust evidence for the effect on protein lipidation, we therefore performed palmitoyl-proteome profiling in KPCA-C cells treated with **3a** using LC-MS/MS[57] (Supplementary Data 1), which confirmed global reductions in palmitoylation. Glycosylation analysis revealed a global decrease in O-GlcNAcylation, while total sialylation levels appeared unaffected (Fig. 5b). However, imaging showed a redistribution of sialylated proteins from the perinuclear region to the cytosol upon treatment with **3a** (Fig. 5c), indicating a disruption of sialylation, even though overall fluorescence intensity remained unchanged (Fig. 5b). A detailed analysis of LC-MS/MS results (Fig. 5d) reveals several Ras isoforms (HRas, NRas, RRas), as key oncogenic GTPases, were among the affected proteins, their oncogenic functions likely being impaired as a consequence of defective lipidation[60]. Lamtor1, regulating mTORC1 signaling and cell death[61] through palmitoylation, was also affected. These results confirmed that CyMA-d can impair essential PTM of functionally critical proteins.

## CyMA-d mislocalize proteins

To assess whether CyMA-d affect protein localization, we examined a panel of membrane-associated proteins, including receptor tyrosine kinases (RTKs: EGFR, INSR, VEGFR2, FGFR2), the G protein-coupled receptor (GPCR: CXCR4), and G-proteins (RAS, GNAQ) across the cell lines that expressed high levels of these proteins (Fig. 5e). After **3a** treatment, EGFR, INSR and VEGFR2 relocated from the PM to the perinuclear region, whereas FGFR2, normally at the Golgi, dispersed into the cytoplasm. Unlike ligand- or inhibitor-induced internalization, which follows regulated endocytosis[62], the redistribution observed here likely results from disrupted intracellular trafficking. For example, syntaxin-6, which associates with a variety of SNARE proteins[63] and mediates RTK translocation at the Golgi[64], was also perturbed following the treatment by **3a** (Supplementary Fig. 9a). Similarly, CXCR4[65] and the GTPase RAS[66] relocated to the perinuclear region, and GNAQ no longer localized at the Golgi after treatment. These results demonstrate that Golgi disruption by CyMA-d broadly affects protein compartmentalization.

## CyMA-d impair protein secretion

Besides protein localization, we evaluated whether CyMA-d influence the secretory function of cells because CyMA-d disrupt anterograde

protein trafficking. Secretion of the immunosuppressive cytokine TGF-β1 was measured in conditioned media from cancer-associated fibroblasts (CAFs), senescent tumor cells (STCs) (Supplementary Fig. 9b), and cancer cell lines treated with **3a** (500 nM) or vehicle (DMSO). In all cases, TGF-β1 secretion was markedly reduced within 4 h of treatment (Fig. 5f). Similarly, VEGF, a key cytokine involved in tumor angiogenesis, also exhibited reduced secretion in response to CyMA-d (Fig. 5g). These findings suggest that Golgi-targeted disruption by CyMA-d compromises the secretory pathway, with potential implications for tumor progression and immune modulation.

## CyMA-d inhibit cell proliferation

We tested **3a** across a wide range of human and mouse cell lines (Supplementary materials). Our results revealed high cytotoxicity against most ovarian cancer cell lines, as well as H460, Saos-2, A431, B16F10, and HeLa cells, with $GI_{50}$ values around 500 nM after 24-hour treatment (Fig. 5h and Supplementary Fig. 10a). The $GI_{90}$ values of **3a** for A431, B16F10, SKOV-3, KPCA-A, KPCA-B, KPCA-C, FT33, FT33 + RAS, FT33 + MYC, FT190 + RAS are below 2 μM (1.2 μg/mL) (Fig. 5h), demonstrating high efficiency and translational potentials of CyMA-d. These results suggest that, due to its disruption of Golgi dynamics and function, CyMA-d effectively trigger cell death in various cancer cell lines at low concentrations.

Furthermore, the design of CyMA-d imparts programmable cell selectivity. Certain cell lines, like HepG2 and THP-1, were highly resistant ($GI_{50} > 20$ μM), a trait that correlated with their high expression of CES that compromises the assembly ability of the molecules by hydrolyzing their ester bonds (Fig. 5h and Supplementary Fig. 10b). This was confirmed as inhibiting CES with BNPP re-sensitized resistant cells to **3a**, while an amide-linked analog (**6**), impervious to CES cleavage, was non-selective (Fig. 5i and Supplementary Fig. 10c). This demonstrates that CyMA-d's activity can be rationally tuned based on the enzymatic profile of target cells.

The cytotoxic effect of **3a** was also significantly mitigated by inhibition of zDHHCs using 2-BP or reducing accessible palmitoyl-CoA with triacsin C (Fig. 5j and Supplementary Fig. 11a). Knockdown of several zDHHC isoforms, respectively, also reduces the cytotoxicity of **3a** (Supplementary Fig. 11b). These results indicate that palmitoylation mediated by multiple zDHHCs, as well as the engagement of endogenous palmitoyl-CoA, are critical for CyMA accumulation at the Golgi and its downstream cytotoxic effects. Structural analogs of **3a**, in which the thioester was replaced by a free thiol (**4a**), an ester (**4b**), or a hydroxyl group (**4c**), all exhibited reduced cytotoxicity (Supplementary Fig. 11c), likely due to altered reactivity or inability to participate in the palmitoylation-depalmitoylation cycle. This confirms that cell death is not a passive effect of the molecule itself but is a direct consequence of its enzymatic activation and subsequent self-assembly driven by endogenous cellular resources.

Moreover, CyMA-d overcome a primary failure point of conventional chemotherapy, acquired drug resistance. We challenged KPCA ovarian cancer cells, a model known for therapy resistance[67], with **3a** (Supplementary Fig. 12a). Remarkably, the $GI_{50}$ values of **3a** for non-stimulated KPCA-B and KPCA-C cells are 390 nM and 280 nM, respectively (Fig. 5k and Supplementary Fig. 12b). After stimulation,

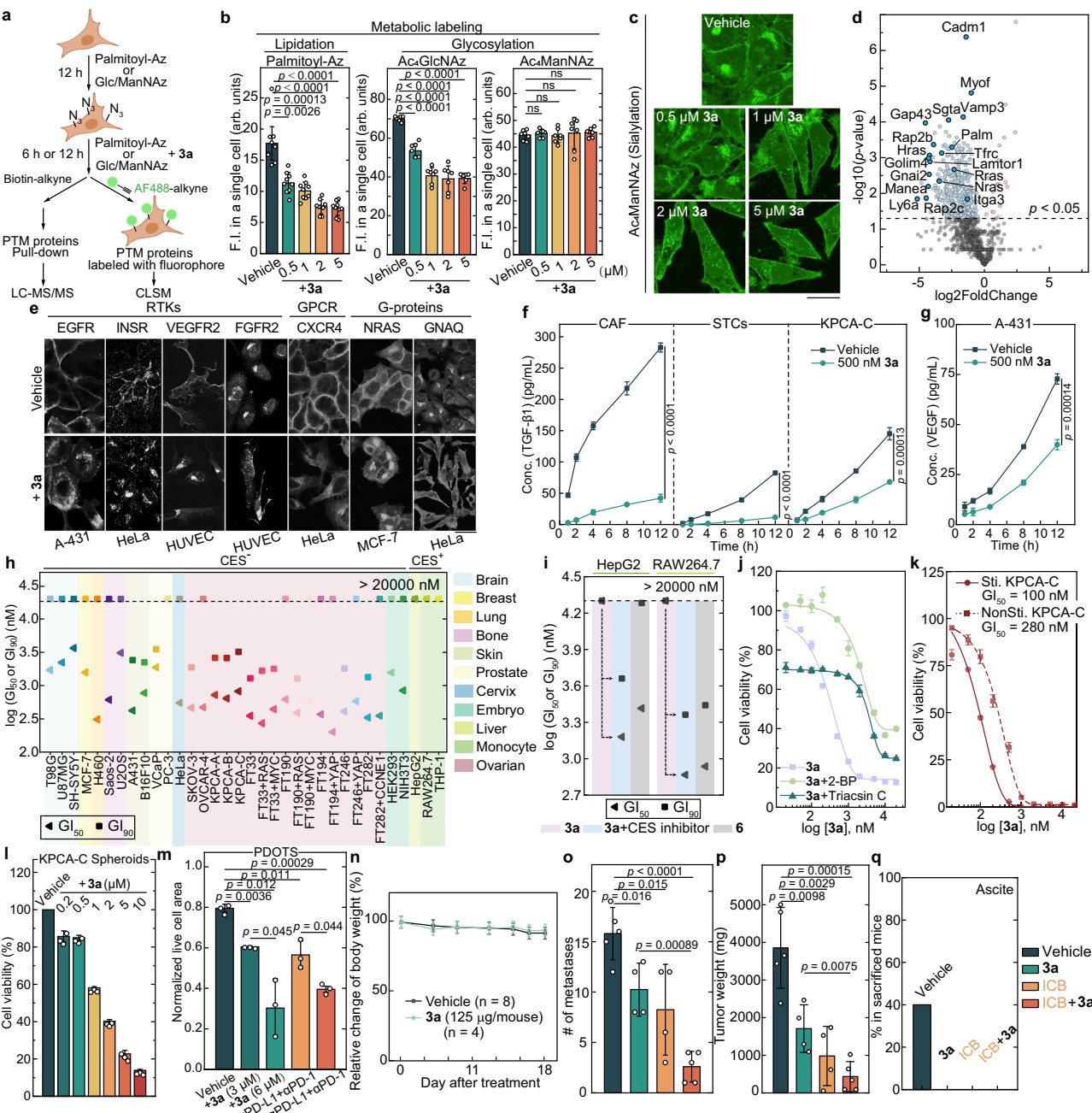

**Fig. 5 | CyMA-d disrupt posttranslational modifications and cytokine secretion and lead to cell death. a** Metabolic labeling scheme for protein palmitoylation or O-GlcNAcylation or sialylation. **b** Protein modification analysis in HeLa treated with **3a** (12 h) via metabolic labeling ($n = 10$ for palmitoylation analysis, and $n = 7$ for O-GlcNAcylation or sialylation analysis). **c** CLSM of protein sialylation in HeLa cells treated with **3a** (12 h). **d** Volcano plot of regulated palmitoylated proteins in KPCA-C cells treated with **3a** (2 µM, 6 h) ($n = 3$). **e** Relocation of representative RTKs, GPCR and G-proteins in different cell lines with/without **3a** (2 µM, 6 h). **f, g** ELISA of TGF-β1 (**f**) and VEGF (**g**) in conditioned media from indicated cells treated with **3a** (500 nM) ($n = 3$). **h** GI$_{50}$ and GI$_{90}$ of **3a** against various cell lines. **i** GI$_{50}$ and GI$_{90}$ of **3a** and **6** in CES-overexpressing cells (HepG2, RAW264.7) ± BNPP (50 µM, 24 h). **j** HeLa viability after **3a** treatment (24 h), with/without 2-BP (10 µM) or triacsin C (10 µM) ($n = 3$).

**k** Viability of stimulated/non-stimulated KPCA-C cells treated with **3a** for 24 hours. **l** Viability of KPCA-C cell spheroids treated with **3a** for 72 hours ($n = 3$). **m** Live/dead staining of PDOTS treated with **3a** ± immune checkpoint inhibitors (200 µg/mL anti-PD-1 and anti-PD-L1) ($n = 3$). **n** Body weight change of mice injected with vehicle or **3a** (125 µg/mouse, three times a week for a total 8 injections) intraperitoneally in PBS ($n = 4$). **o** Number of metastases, (**p**) Tumor weight and (**q**) presence of ascites in KPCA-B bearing mice treated with vehicle, **3a** alone (3.6 µg/mouse), ICB alone (50 µg/mouse anti-PD-L1 and 50 µg/mouse anti-CTLA-4), or combination ($n = 5$ for vehicle and combination, $n = 4$ for the rest). Scale bar = 20 µm. Data are mean ± s.d. ($n$-values as indicated in the panel). Statistical significance was determined by a two-tailed Student's $t$ test. Reproducibility and statistical details are provided in the Methods. Source data are provided as a Source Data file.

these values even decreased to 180 nM and 100 nM, respectively. This sensitization effect highlights a paradigm-shifting advantage of this assembly-based approach, which leverages, rather than being defeated by, the complex cellular state to drive potent antitumoral efficacy.

## Ex vivo and in vivo antitumoral performance of CyMA-d

This principle of Golgi disruption via in situ assembly was effective in more complex biological systems. In 3D KPCA-C spheroids, treatment with **3a** inhibited viability by 80% at a 5 µM concentration (Fig. 5l).

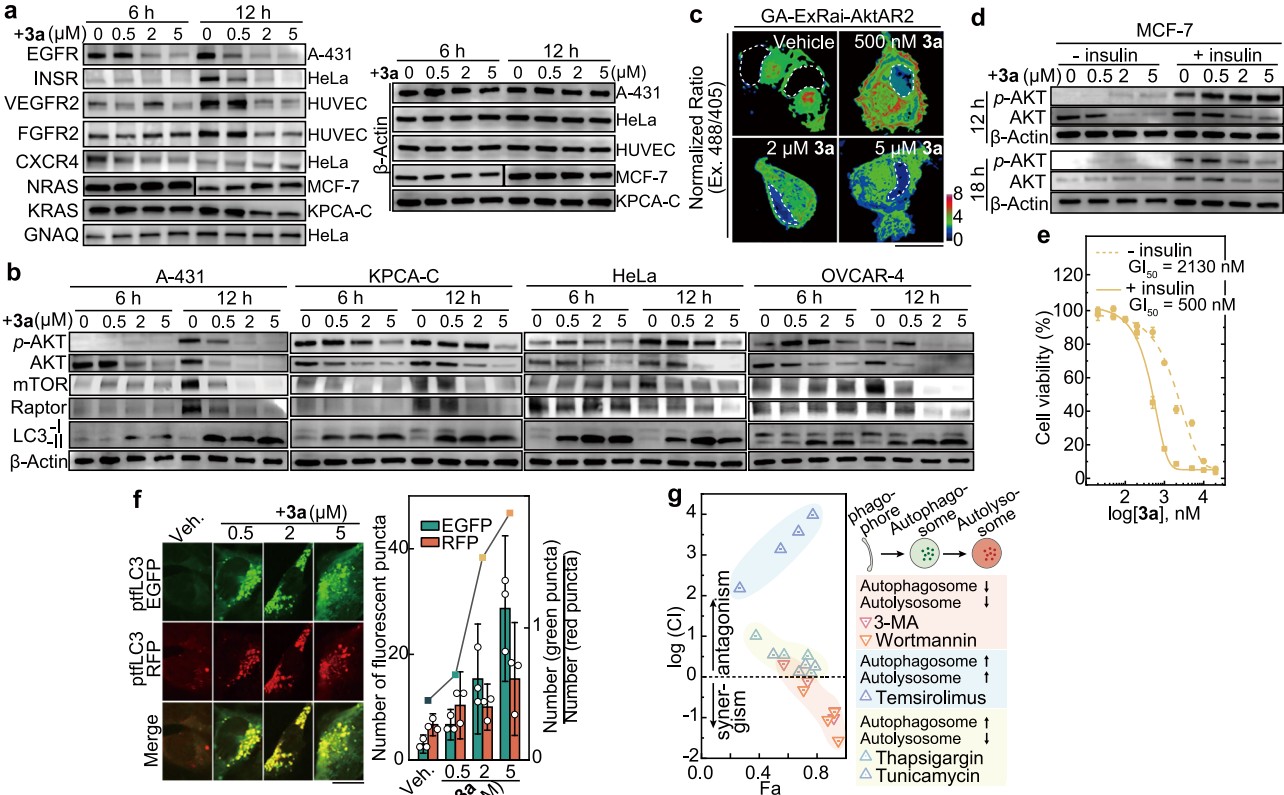

**Fig. 6 | CyMA-d disrupt RTK signaling and lead to cell death. a** Immunoblotting of the representative RTKs (EGFR, INSR, VEGFR2, FGFR2), GPCR (CXCR4) and G-proteins (NRAS, KRAS, GNAQ) in different cell lines with or without the treatment of **3a** (500 nM, 2 µM, 5 µM; 6 and 12 h). **b** Immunoblotting of significant proteins (p-AKT, AKT, mTOR, Raptor, LC3B) in the AKT-mTOR signaling pathway in various cell lines with or without the treatment of **3a** (500 nM, 2 µM, 5 µM; 6 and 12 h). **c** Determination of AKT activity using a fluorescent biosensor, GAExRai-AktAR2. **d** Immunoblotting of p-AKT and AKT in MCF-7 cells stimulated with or without insulin and treated with or without **3a** (500 nM, 2 µM, 5 µM; 12 and 18 h). **e** Cell viability of insulin-stimulated and unstimulated MCF-7 cells treated with or without **3a** for 24 h. **f** CLSM of ptfLC3-HeLa cells treated with or without **3a** (500 nM, 2 µM,

5 µM; 6 h), along with the quantification of green and red fluorescent puncta per cell and the ratio of green to red puncta (n = 3). **g** Quantitative analysis of synergism and antagonism in combinations of **3a** with autophagy regulators. 3-methyladenine (3-MA) and Wortmannin inhibit autophagosome formation and further maturation; Temsirolimus activates autophagosome formation and autolysosome maturation; Thapsigargin and Tunicamycin activate autophagosome formation but block the autolysosome formation. Data are mean ± s.d. (n-values as indicated in the panel). Statistical significance was determined by a two-tailed Student's t test. Reproducibility and statistical details are provided in the Methods. Source data are provided as a Source Data file.

Similarly, in patient-derived organotypic tumor spheroids (PDOTS) model, **3a** induced a significant, concentration-dependent loss of cell viability (Fig. 5m and Supplementary Fig. 12c), confirming effective inhibition of spheroid proliferation. Since we have illustrated that CyMA-d impair the secretion of immunosuppressive cytokines and PDOTS retain key features of the native tumor immune microenvironment[68], we conducted a combination study with CyMA-d and immune checkpoint blockade (ICB; anti-PD-L1 and anti-PD-1) to assess potential synergistic effects. The combination enhanced the inhibition of cell proliferation (Fig. 5m and Supplementary Fig. 12c), suggesting that compromising Golgi function can favorably alter the tumor microenvironment for inhibiting tumor growth.

We evaluated the consequences of this mechanism in a syngeneic in vivo tumor model (Supplementary Fig. 12d). Intraperitoneal administration of **3a** was well-tolerated (Fig. 5n). Even at a low dose (0.144 mg/kg), the intraperitoneal administration of CyMA-d led to a significant reduction in metastases, tumor weight, and ascites incidence (Fig. 5o–q). Consistent with the PDOTS data, combining **3a** with ICB produced a synergistic effect, leading to superior tumor control and a 100% survival rate (Supplementary Fig. 12e). These results demonstrate that the targeted, intracellular generation of a disruptive material can be translated from cell lines to in vivo systems, providing a powerful supramolecular strategy for controlling tumor growth by compromising a fundamental organelle.

## Mechanistic studies of cell death triggered by CyMA-d

The formation of CyMA-d assemblies within the Golgi initiates a catastrophic failure in cellular protein trafficking, leading to a cascade of downstream effects. A primary consequence is the profound disruption of RTK homeostasis. We observed a significant reduction in the total levels of key RTKs, including EGFR, INSR, VEGFR2, and FGFR2, following treatment with **3a** (Fig. 6a). This is not simple downregulation but rather a consequence of disrupting the critical sorting and transport functions of the Golgi, trapping proteins and preventing their proper localization, which ultimately leads to their degradation.

RTKs are pivotal in cancer pathogenesis[69], including their influence on the AKT signaling pathway, which regulates key processes like cell growth, survival, and proliferation[70]. We examined proteins in this pathway across several cancer cell lines (A-431, KPCA-C, HeLa, OVCAR-4) (Fig. 6b) and in NIH3T3 embryonic fibroblasts (Supplementary Fig. 13a). Our data showed a consistent, concentration-dependent downregulation of both AKT and phosphorylated AKT in all four cell lines treated with **3a**. Moreover, the decreased mTOR and Raptor expression suggests a reduced mTORC1 activity in these cell lines. Using an excitation-ratiometric assay, we assessed whether CyMA-d affects AKT kinase activity (Fig. 6c)[71]. Reduced normalized fluorescent intensity ratios at the two excitation wavelengths (Ex 488/405) in treated cells confirmed that CyMA-d decrease both AKT expression

and its kinase activity. Collectively, these results indicate that CyMA-d consistently disrupt RTK/AKT pathway.

We also investigated the RAS/ERK/AKT pathway by analyzing ERK, P38, and JNK phosphorylation using immunoblotting (Supplementary Fig. 13b). Phosphorylation levels varied across cell lines, with ERK1/2 upregulated in KPCA-C but downregulated in OVCAR-4, likely reflecting differences in RAS expressions and dependence among these lines.

To validate CyMA-d disrupting RTK/AKT signaling as a mechanism for inhibiting cell growth, we cultured MCF-7 cells with insulin to activate the AKT pathway[72,73] and assessed how CyMA-d affected cells with or without insulin stimulation (Fig. 6d, Supplementary Fig. 13c). Phosphorylated AKT level significantly increased in insulin-stimulated MCF-7 cells, confirming pathway activation. A 12-hour 3a treatment did not significantly alter phosphorylated AKT in either insulin-stimulated or unstimulated MCF-7 cells, although total AKT levels decreased in both. In contrast, after 18 h, phosphorylated AKT decreased in insulin-stimulated cells, while remaining unchanged in unstimulated cells (Fig. 6d). We found that insulin-stimulated MCF-7 cells were more sensitive to 3a, with an GI$_{50}$ of approximately 500 nM compared to about 2.1 μM for unstimulated cells (Fig. 6e). These results suggest that AKT pathway activation and dependence sensitizes cells to CyMA-d, confirming that AKT signaling disruption is one of the key mechanisms of CyMA-d-induced cell death.

An increased LC3-II/LC3-I ratio (Fig. 6b) suggests enhanced autophagosome formation[74], supported by immunocytochemistry (Supplementary Fig. 14a). Employing tandem fluorescent assay[75], we confirmed that CyMA-d might arrest autophagosome maturation (Fig. 6f). This distinctive property of CyMA-d promoted us to explore its relationship with autophagy-induced cell death. We co-incubated cells with CyMA-d and various autophagy regulators, followed by analysis by CompuSyn[76]. As shown in Figs. 6g, 3a demonstrated clear synergism with autophagy inhibitors (3-MA and wortmannin), which block autophagosome formation and further maturation[77]. Conversely, 3a exhibited antagonism with the autophagy activator temsirolimus[78], and weaker antagonism with thapsigargin and tunicamycin, which promote autophagosome formation but inhibit maturation into autolysosomes[79]. These findings suggest that CyMA-d-induced autophagosome formation initially acts as a survival mechanism under severe intracellular stress[80], yet inhibition of subsequent maturation leads to cell death. A more condensed pattern of LC3 induced by CyMA-d, compared to chloroquine, suggests a unique mechanism of autolysosome inhibition (Supplementary Fig. 14a). This is further evident as PI4KIIα, crucial for autophagosome-lysosome fusion and dependent on palmitoylation[81,82], dispersed into small cytoplasmic puncta after 3a treatment (5 μM) (Supplementary Fig. 14b) due to disrupted palmitoylation[82]. These findings underscore the intricate role of autophagy in CyMA-induced cell death[83].

CyMA-d treatment also triggered an extensive ubiquitin-proteasome response, marked by a threefold increase in poly-ubiquitinated proteins, including S-acylated ones like integrins, which links the disruption of palmitoylation to a systemic protein degradation crisis (Supplementary Fig. 14c–e and Supplementary Data 2).

All the results indicate that CyMA-d do not act as conventional inhibitors but as in situ-generated assemblies that physically obstruct Golgi function. This single perturbation triggers a multi-pronged, cascading collapse of essential cellular processes, from signaling to degradation, which proves irreversibly fatal. This mechanistic complexity explains why inhibitors of any single downstream pathway fail to rescue the cells (Supplementary Fig. 14 f), highlighting a unique and powerful consequence of intracellular assembly formation.

## Discussion

This study demonstrates CyMA as a first-in-kind supramolecular entity generated in situ for pleiotropic Golgi-targeting in a cell-selective manner. Although acylation of certain proteins or specific peptide sequences have been previously demonstrated[42,84,85], this study marks a instance where reversible acylation of ultrashort peptides in cells produces dynamic and non-diffusive supramolecular assemblies. Unlike conventional peptide amphiphile assemblies that are pre-formed before interacting with cells[86] or organelle-selective lipid labeling strategies[87], CyMA are assembled locally by hijacking the endogenous enzymatic machinery to conjugate a small molecule precursor (< 700 Da) with endogenous palmitoyl-CoA, meaning the cell itself is forced to supply roughly one-third of the final assemblies' mass, a strategy that maximizes efficacy in triggering a cellular response.

Contrasting to reversible gelation of small molecule assemblies by in vitro chemical reactions[88], this work shows in cellulo transient molecular assemblies, promising broad applications. The dynamic thioester bond formation and breaking[89] endows CyMA with its unique cycling nature, which allows for high efficacy and requires considerably fewer precursors to achieve comparable results to supramolecular assemblies formed by the catalysis of only one kind of enzymes[90]. This efficiency is crucial for its dual roles, as a minimally-impact imaging tool (CyMA-i) and as a potent modulator of cell function (CyMA-d).

PPT1's documented localization at the Golgi[91,92] and its ability to retain enzymatic activity at pH > 4.5[93,94] support a model in which CyMA undergoe local, in situ deacylation, enabling dynamic cycling at the Golgi. This localized enzymatic turnover drives a continuous assembly–disassembly loop and likely operates in parallel with cytosolic thioesterases such as LYPLA1/2, whose individual contributions we were unable to exclude due to inhibitor cytotoxicity. Importantly, these depalmitoylation pathways are not mutually exclusive. Even CyMA molecules that are depalmitoylated by PPT1 in lysosomes can return to the Golgi through extensive vesicular and non-vesicular lipid exchange[95], where they can be repalmitoylated. Likewise, other depal-mitoylases, such as members of the ABHD family[96], may also contribute to CyMA cycling. Together, these routes reinforce a robust palmitoylation–depalmitoylation cycling mechanism. Furthermore, our chase experiments using a super-sensitive and ultrafast Golgi probe[97] provide additional support for this dynamic cycling model.

The accumulation of CyMA-d assemblies physically disrupts essential Golgi functions. While the non-fluorescent CyMA-d probe cannot be directly visualized and could in principle localize to other membranes where palmitoylation occurs, several evidences support that Golgi disruption results in the observed phenotypes. First, the fluorescent version of CyMA-d confirmed direct targeting by selectively accumulating in the Golgi. Second, Golgi's role as the central hub for S-palmitoylation[7] makes it the most favorable site for CyMA's activity. Third, the phenotypes follow a clear cascade where Golgi damage impairs trafficking and downstream signaling. Although contributions from other organelles cannot be ruled out, the available evidence is most consistent with Golgi disruption as the predominant effect. More than simple organelle disruption, this process culminates in a profound reorganization of Golgi-derived membranes into distinct, aberrant subcellular structures (Supplementary Fig. 15). Our results suggest that CyMA-d act as both a substrate for palmitoylation and, over time, a disruptor of the process. In the early stages, 3a engages the palmitoylation/depalmitoylation cycle to accumulate at the Golgi, while the global reduction in endogenous palmitoylation is a downstream effect of this disruption. As a small molecule, CyMA is expected to undergo faster kinetics than proteins, enabling a dynamic turnover where it can transiently detach via depalmitoylation. Its effective lifetime at the Golgi, therefore, reflects the balance between detachment and the now-impaired re-palmitoylation on these disrupted membranes. These aberrant Golgi-derived structures likely serve as the main sites of this turnover, and determining the precise lifetime of the probe within them will be a focus of future work. The sustained presence of these active de novo compartments, which are

dynamically maintained by this futile cycle, appears to be a critical mechanistic event, likely driving cytotoxicity through complex interactions with intrinsic cellular machinery. While the precise nature of these interactions is the focus of ongoing investigations, the formation of these structures provides a compelling physical basis for the potent, multi-pathway disruption that canonical inhibitors for apoptosis, necroptosis, or ferroptosis are unable to rescue (Supplementary Fig. 14f).

This unique mechanism of action provides a fundamentally different paradigm from conventional inhibitors and is what endows these intracellular materials with significant translational value. While traditional therapeutic strategies often rely on specific binding, they are vulnerable to resistance[98]. In contrast, CyMA-d leverage a well-conserved enzymatic process for its activation. The reversibility of S-palmitoylation is likely what drives the strong phenotype. The continuous palmitoylation/depalmitoylation creates a futile cycle that persistently consumes palmitoyl-CoA, placing a sustained metabolic burden on the cell. At the same time, this cycling dynamically maintains the solid-like assemblies at the Golgi, which physically obstruct its function, severely impairing trafficking and signaling. Thus, the potent cellular effects arise not despite the reversibility, but because of it; the combination of metabolic drain and physical obstruction overwhelms cellular homeostasis, ultimately leading to cell death. Since different signaling pathways often undergo genetic alterations across cancer types, CyMA-d inhibit cell proliferation by disrupting both RTK/AKT and RAS/ERK/AKT signaling pathways, acting differently from the conventional RTK or RAS inhibitors. This pleiotropic action is particularly effective in complex disease states, as evidenced by the heightened sensitivity of RAS-driven cancer cells to CyMA-d (Fig. 5h). Such sensitivity likely reflects the fundamental dependence of RAS oncoproteins on post-translational lipidation for their function[42,99]. Proper membrane localization and signaling of RAS isoforms require S-palmitoylation at the Golgi; therefore, the disruption of Golgi integrity and the impairment of global S-palmitoylation by CyMA-d likely interfere with RAS processing and membrane anchoring, leading to the collapse of downstream survival pathways in these RAS-addicted cancer cells.

Furthermore, by compromising the central hub of the secretory pathway, CyMA-d assemblies offer a powerful method to modulate the cellular secretome. This approach concurrently blocks the secretion of signaling molecules (e.g., VEGF, TGF-β) while also preventing the proper cell surface localization of their receptors (e.g., VEGFR2). This dual disruption dismantles entire intercellular communication loops, a strategy that is inherently more robust against the development of resistance in cancer therapy.

Looking forward, the concept of establishing enzyme-controlled, futile assembly-disassembly cycles at defined subcellular locations and consuming endogenous metabolites provides a broadly applicable framework. To realize the full potential of this approach, several critical next steps are required. These include characterizing the precise nature of the observed subcellular reorganizations and their cargo, as well as undertaking a detailed quantitative analysis of this dynamic system. Furthermore, a detailed investigation into the various upstream endocytic pathways and specific proteins most affected by CyMA-d using genetic approaches would also be a valuable future direction. By moving beyond specific binding and leveraging the cell's own machinery to build functional materials in situ, this work opens unexplored directions for biological discovery and for targeting conserved cellular processes in disease. Our division of the platform into CyMA-i for imaging and CyMA-d for disruption is a first step toward ensuring these tools are used with precise intent. To further enhance this specificity, we are actively developing next-generation CyMA variants designed to be more sensitive and minimally disruptive, so that the imaging and perturbation functions can be better separated, which will further strengthen the utility and precision of the platform in future applications.

## Methods

### Ethical statement
All research described in this study complies with all relevant ethical regulations. All animal procedures and experimental protocols involving the syngeneic mouse tumor models (KPCA-B) were approved by the Institutional Animal Care and Use Committee (IACUC) of Brigham and Women's Hospital. We confirmed that the maximal permitted tumor burden was not exceeded in any experimental animals during the study. Mice were monitored for body weight changes three times per week for the duration of the treatment to ensure compliance with these ethical limits. For the studies involving patient-derived organotypic tumor spheroids (PDOTS), the study protocol was reviewed and approved by the Institutional Review Board (IRB) of Brigham and Women's Hospital.

### Cell culture
T98G (CRL-1690), U87MG (HTB-14), SH-SY5Y (CRL-2266), MCF-7 (HTB-22), H-460 (HTB-177), U2OS (HTB-96), A-431 (CRL-1555), B16F10 (CRL-6475), VCaP (CRL-2876), PC-3 (CRL-1435), HeLa (CCL-2), HEK293 (CRL-1573), NIH3T3 (CRL-1658), HepG2 (HB-8065), RAW264.7 (TIB-71), THP-1 (TIB-202), SKOV-3 (HTB-77), hTERT PF179T CAF (CRL-3290) cells were purchased from ATCC. OVCAR-4 (SCC258) was purchased from Sigma. KPCA-A, KPCA-B and KPCA-C cells were provided by Dr. Daniela Dinulescu lab. Saos-2 cells were provided by Prof. David Loeb lab. FT33, FT33 + RAS, FT33 + MYC, FT190, FT190 + RAS, FT194, FT194 + YAP, FT246, FT246 + YAP, FT282, FT282 + CCNE1 were provided by Dr. Ronny Drapkin lab. Cell lines were authenticated by CellCheck 9 - human (9 Marker STR Profile and Inter-species Contamination Test, IDEXX), confirming 100% match of the cell identity. T98G, MCF-7, HeLa, HEK293, HepG-2 cells were cultured in MEM supplemented with 10% FBS. SH-SY5Y cells were cultured in 1:1 mixture of EMEM and F12 Medium supplemented with 10% FBS. A-431, B16F10, VCaP, NIH3T3, RAW264.7 cells were cultured in DMEM supplemented with 10% FBS. H-460, Saos-2, THP-1, OVCAR-4 cells were cultured in RPMI 1640 medium supplemented with 10% FBS. U2OS, SKOV-3 cells were cultured in McCoy's 5 A medium supplemented with 10% FBS. U-87 MG cells were cultured in EMEM supplemented with 10% FBS. MCF-7 cells were cultured in EMEM supplemented with 10% FBS and 0.01 mg/mL human recombinant insulin. PC-3 cells were cultured in F-12K supplemented with 10% FBS. hTERT PF179T CAF cells were cultured in EMEM supplemented with 10% FBS and 1 μg/mL puromycin. KPCA-A, KPCA-B and KPCA-C cells were cultured in mFT cell media. FT33, FT33 + RAS, FT33 + MYC, FT190, FT190 + RAS, FT194, FT194 + YAP, FT246, FT246 + YAP, FT282, FT282 + CCNE1 cells were cultured in DMEM/F12 50:50 Mix without L-glutamine, supplemented with 10% FBS. All the cell lines were supplemented with 100 U/mL penicillin and 100 μg/mL streptomycin and were cultured and humidified with 5% $CO_2$ at 37 °C.

### Cell lines used to test cytotoxicity of CyMA-d 3a
We tested **3a** across a wide range of human and mouse cell lines, including brain (T98G, U87MG, SH-SY5Y), breast (MCF-7), lung (H-460), bone (Saos-2, U2OS), skin cancer / melanoma (A431, B16F10), prostate (VCaP, PC-3), cervix (HeLa) cancer cell lines, embryonic-derived cell lines (HEK293, NIH3T3), hepatocytes (HepG2), monocytes (RAW264.7, THP-1), and ovarian cancer cell lines (SKOV-3, OVCAR-4, KPCA-A, KPCA-B, KPCA-C, FT33, FT33 + RAS, FT33 + MYC, FT190, FT190 + RAS, FT190 + MYC, FT194, FT194 + YAP, FT246, FT246 + YAP, FT282, FT282 + CCNE1).

### Cell transfection
**siRNA transfection.** Cells were seeded at $2 \times 10^5$ cells per well on the 6-well plate for 24 h to allow attachment. Upon reaching 60-70% confluency, the cells were transfected with Lipofectamine 3000 reagent. Specifically, 5 μL of Lipofectamine 3000 and 5 μL of siRNA stock solution (20 μM) were separately dissolved in 250 μL of Opti-

MEM and briefly vortexed before mixing. The mixture solution was left to stand for 15 min. Meanwhile, the media in each well was rinsed and replaced with 1.5 mL of Opti-MEM. The Lipofectamine 3000 and siRNA mixture was added dropwise to each well and incubated for 6 hours at 37 °C, after which the Opti-MEM was replaced with fresh culture media containing FBS and P/S. The cells were then incubated for 48 h before proceeding with live-cell imaging or cell lysis for further analysis.

### Plasmid transfection
Cells were seeded at $1.5 \times 10^5$ cells per confocal dish for 24 h to allow attachment. Upon reaching 40–50% confluency, the cells were transfected with Xfect™ Transfection Reagent. Specifically, 5 µg of the plasmid DNA was diluted with Xfect Reaction Buffer, followed by the addition of 1.5 µL Xfect Polymer. The mixture was briefly vortexed and incubated for 10 min at room temperature. The entire 100 µL of nanoparticle complex solution was then added dropwise to the cell culture medium, and the dish was rocked briefly. The confocal dish was incubated at 37 °C overnight, and the media was replaced with fresh culture media for an additional 48-hour incubation. The cells were then ready for live-cell imaging.

### Confocal microscopy
A confocal dish (35 mm dish with 20 mm bottom well, #1.5 glass) was used to prepare CLSM samples. For live-cell imaging, cells in the exponential growth phase were seeded on the confocal dish at $1.0 \times 10^5$ cells per dish and incubated for 24 h. After removing the culture medium, fresh medium containing the compound of interest was added to the cells for co-incubation at 37 °C in a humidified atmosphere of 5% $CO_2$ for the desired period. Afterwards, the nuclei of cells were stained with Hoechst 33342 for 10 min, and the samples were washed with 1 mL of Live Cell Imaging Solution four times to fully remove the residual Hoechst 33342.

For time-lapse live-cell imaging, cells in the exponential growth phase were seeded on a confocal dish at $1.0 \times 10^5$ cells per dish and incubated for 24 h. The samples were washed with 1 mL of Live Cell Imaging Solution three times, and the nuclei were stained with Hoechst 33342 for 10 min. The samples were then washed with 1 mL of Live Cell Imaging Solution four times to remove the residual Hoechst 33342. The position of cells and the focal plane of the laser beam were determined using the fluorescence from the stained nuclei with a laser with a 405 nm wavelength, and the Nikon Perfect Focus System was activated to prevent focus drift. The imaging solution in the confocal dish was replaced with fresh imaging solution containing the compound of interest. CLSM images of different channels were then recorded, with the time-series interval set to be 1 minute or no delay. After a specified number of imaging cycles, the fluorescence images from both channels were saved for further analysis.

### Colocalization study with GALNT2-RFP
HeLa cells ($1.5 \times 10^5$ cells) were seeded in a confocal dish for 24 h to allow attachment. The culture media was then replaced with fresh medium containing CellLight™ Golgi-RFP, BacMam 2.0 at the concentration of 2 µL per 10,000 cells, and the cells were incubated for another 24 h to complete the transduction. The medium was then replaced with fresh media containing **1a** (1 µM) and incubated for 10 minutes for live-cell imaging. The Pearson's R value was calculated using the Coloc 2 plugin in Fiji.

### Cell pretreated with inhibitors
HeLa cells ($1.5 \times 10^5$ cells) were seeded in a confocal dish for 24 h to allow attachment. The culture media was then replaced with fresh medium containing endocytosis inhibitors (mβCD, EIPA, CPZ, Dynasore, 7-keto-chol), or inhibitors for LYPLA1/2 (ML211, 50 µM), PPT1 (DC661, 20 µM), palmitoylacyltransferases (2-BP, 50 µM), CES1 (Nevadensin, 20 µM), or CES2 inhibitor (Loperamide, 20 µM), and the cells

were incubated for 30 minutes at 37 °C. Afterward, the cell medium was replaced with fresh medium containing CyMA for live-cell imaging.

### FRAP assay
FRAP was performed on a Zeiss LSM 880 confocal microscopy using a $63 \times /1.4$ Oil objective or a Nikon AX-R resonant confocal system using a $60 \times /1.4$ Oil objective. Five pre-bleach images were captured, followed by photobleaching using 488 nm laser at 100% intensity within a selected region. Another region was imaged without photobleaching as an internal control. $512 \times 512$-pixel images were captured at 0.26 s (or 5.02 s for Nikon AX-R CLSM) intervals using a 488 nm laser at 100% intensity with the pinhole set at 1 airy unit. Imaging continued until no further recovery was observed. The fluorescence recovery of the photobleached region was normalized and fitted into an exponential function.

### Structured illumination microscopy (SIM) imaging
A 3D-Nikon structured illumination microscopy (N-SIM, version AR5.11.00 64 bit, Tokyo, Japan), equipped with solid-state lasers (488 nm, 561 nm, 640 nm, the output powers at the fiber end: 15 mW) and an Apochromat $100 \times /1.49$ numerical aperture oil-immersion objective lens, was used to acquire all SIM images. Images were obtained using Nikon NIS-Elements $512 \times 512$ resolution, with Z-stacks. NIS-Elements AR Analysis was used to reconstruct and process raw images.

Cells were seeded on glass-bottomed culture dishes (MatTek; P35G-1.5-14-C) for 24 hours to allow for adhesion. For Golgi staining, cells were treated with **1a** (2 µM) for 5 min. Before imaging, cells were washed with PBS three times. Green channel images (emission bandwidth: 500–550 nm) were excited with a 488 nm laser, and red channel images (emission bandwidth: 570–640 nm) were excited with a 561 nm laser. Imaging data analysis was performed using ImageJ. Mitochondrial and ER analysis were conducted as per previously reported references[100,101].

### ER-to-Golgi anterograde trafficking analysis
HeLa cells seeded on confocal dishes were transfected with various RUSH plasmids. The cells were then treated with CyMA for 6 h at 37 °C, followed by nuclear staining with Hoechst 33342. The cells were transferred to the CLSM for imaging, and the medium was replaced with fresh medium containing 40 µM of biotin. Imaging began immediately after the addition of biotin using time-series mode. CLSM images were saved for further analysis.

### Golgi-to-PM anterograde trafficking analysis
HeLa cells seeded on the confocal dishes were transfected with various RUSH plasmids. Afterward, cells were treated with 40 µM of biotin for 1 hour at 20 °C to allow cargo proteins to accumulate at the Golgi but inhibit their subsequent trafficking to the plasma membrane. Cells were then treated with CyMA for 6 hours at 20 °C. The cells were transferred to CLSM sites, and the temperature was raised to 37 °C. Imaging of the cells was performed immediately after increasing the temperature using time-series mode. The CLSM images were saved for further analysis.

### PM-to-Golgi retrograde trafficking analysis
Giantin-mNeonGreen transfected HeLa cells seeded on the confocal dishes were treated with CyMA at different concentrations for 6 h at 37 °C. Then, the media was replaced with fresh media containing 1 µg/mL CTxB-AF647 and incubated for another 1 h. The cells were sent for CLSM imaging to study the co-localization of CTxB-AF647 with Giantin to evaluate the PM-to-Golgi trafficking behavior.

### Golgi-to-ER retrograde trafficking analysis
Golgi-to-ER trafficking was evaluated by photobleaching the ER pool of GalT and monitoring fluorescence recovery at the ER as an indicator of

retrograde transport from the Golgi to the ER. HeLa cells transfected with GalT-EGFP were seeded in confocal dishes and incubated with CyMA for 6 hours at 37 °C. Afterward, FRAP of the ER pool of GalT was performed as described above. Time-series fluorescence intensity recoveries at the photobleaching sites were recorded for further analysis.

## Immunocytochemistry
The culture media was removed from cells, followed by two washes with PBS. Cells were fixed with 4% paraformaldehyde for 10 min and permeabilized with 0.1% Triton X-100 in PBS for 6 min. Cells were then blocked with 3% BSA and 22.5 mg/mL of glycine in PBST for 1 h at room temperature. Primary antibodies were diluted 1:200 in 1% BSA in PBS and incubated with the cells overnight at 4 °C. Cells were then incubated with Alexa Fluor 647-conjugated secondary antibodies (1:1000 dilution) for 1 h at room temperature. Between each step, except after blocking, cells were washed three times with PBS. The cells were then ready for CLSM imaging.

## Quantification of fluorescence intensity at Golgi
Image processing was conducted to extract single-cell responses. The image pixel values are scaled to be between [0, 1] with 1 indicates the original intensity of 255.
  a. Nuclei Detection. The Otsu method[102] was used on nucleus-staining frames to automatically determine an intensity threshold for detecting foreground pixels corresponding to nuclei. The minimum threshold was set to 0.6 out of 1.0. Morphological operations (open, fill, and erode) were applied to remove noise, fill the holes in the foreground segments, and separate foreground pixels into segments corresponding to individual nuclei. Some nuclei near image boundaries were discarded.
  b. Reaction Signal Detection. The last reaction-staining frame of a video was used to determine the intensity threshold for detecting reaction signals in the entire video. The threshold was set to a value higher than 99% of the pixels in the last frame. Small foreground segments containing less than 4 pixels were considered noise and removed. A foreground segment was considered part of a cell if its corresponding nucleus were close to it.
  c. Golgi Detection. A manual threshold of 0.05 was set to determine the foreground mask representing Golgi signals. Using the Golgi mask, reaction signals were classified as occurring inside or outside the Golgi.

## Palmitoylation of CyMA characterized by LC/HR-MS
A total of $1.2 \times 10^7$ cells were treated with 1 μM of CyMA or vehicle (DMSO) for 30 minutes, and then washed with HEPES buffer twice. Cells were collected and centrifuged to obtain a pellet. The pellet was resuspended in 500 μL of HEPES buffer, followed by the addition of 1.5 mL of DCM and 1 mL of methanol. The mixture was allowed to sit for 10 minutes. Afterward, 0.5 mL DCM and 0.5 mL Tris-HCl (50 mM, pH 2.0) were added, and the tube was centrifuged to collect the organic phase. The organic phase was washed with buffer (1 mL methanol+1 mL Tris-HCl (50 mM, pH 2.0)) and centrifuged again to obtain the organic phase. The organic solvent was evaporated with $N_2$, and the remaining solid was dissolved in 150 μL of methanol and sent for LC/HR-MS analysis. The analysis was performed on a C18 reverse phase column using water (0.1% formic acid) and acetonitrile (0.1% formic acid) as the mobile phases, with a gradient of 1% to 99% acetonitrile over 14 min. The identity of each compound was confirmed by matching the experimentally observed high-resolution mass-to-charge ratio (m/z) with the calculated theoretical exact mass. We further verified these assignments by confirming that the observed isotopic distribution pattern of the peaks in the mass spectrum matched the theoretical pattern predicted for the compound's elemental formula. Experiments were performed in $n = 3$ independent biological replicates.

## *Drosophila* culture, dissection, and imaging
Flies were cultured using standard media and techniques and maintained at 25 °C. The following strains were used and obtained from the Bloomington *Drosophila* Stock Center: He-Gal4 (RRID:BDSC_8699), UAS-ManII-TagRFP (RRID:BDSC_65249), Df(2 R)Exel6078 (RRID:BDSC_7558). Other stocks used include *vps35*[e42 103], C57-Gal4[104], and w1118 [105]. For Hemocyte isolation, the cuticles of 3 wandering third instar larvae were ripped open at the midpoint, avoiding damage to the guts and the hemocytes were released directly into 50 μL M1 medium supplemented with BSA (1.5 mg/mL) and D-glucose (2 mg/mL)[106]. Hemocytes were then plated on coverslip-bottom chamber slides and allowed to settle for 5 min. After settling, the hemocytes were treated with 5 μM **1a** and imaged immediately and after 10 min using Zen Blue software on a Zeiss LSM880 Fast Airyscan microscope in super-resolution acquisition mode using a 63X (n.a. 1.4) oil immersion objective. For imaging of other larval tissues, wandering third instar larvae were pinned down and filleted one at a time in HL3.1 on slides containing sylgard in a silicone mold. The larvae were treated for 10 min in HL3.1 containing 10 μM **1a**. Following treatment, the larvae were quickly washed 2 times in HL3.1, then 50 μL of HL3.1 was added to the larva and covered with a coverslip for imaging. Z-stacks were acquired using a Nikon Ni-E upright microscope equipped with a Yokogawa CSU-W1 spinning disk head, an Andor iXon 897U EMCCD camera, and Nikon Elements AR software. A 60X (n.a. 1.4) oil immersion objective was used to image the m4 muscle, the ventral nerve cord, or the salivary glands.

## MTT assay
The MTT assay was used to determine cell viability for cytotoxicity evaluation. Cells were seeded at $1 \times 10^4$ cells per well in 96-well plates for 24 h to allow attachment. Culture media were replaced with fresh culture media containing the compounds at a series of concentrations. After 24, 48, and 72 h, 10 μL of MTT solution (5 mg/mL) was added to each well, and the plate was incubated in the dark for 4 h at 37 °C. Then, 100 μL of 10% SDS-HCl was added to stop the reaction and dissolve the formazan. The absorbance at 595 nm was determined by a microplate reader. The assay was repeated three times, and the mean values of three measurements were plotted, with error bars representing standard deviation. For 7-day cytotoxicity, seed cells at 5000 cells per well, and fresh D-peptide-containing medium was added every 3 days.

## Immunoblotting
Cells were cultured to 80-90% confluency, lysed with 500 μL lysis buffer (containing protease inhibitor cocktail and phosphatase inhibitor cocktails) per 10 cm dish on ice, sonicated for 10 s, and subjected to three freeze-thawed cycles to collect cell lysates from various cell lines. The lysates were centrifuged at 12,000 rpm for 10 min at 4 °C, and the supernatant was collected. The proteins in the lysates were denatured by adding sample loading buffer and incubating at 95 °C for 8 min. The lysate samples were loaded onto precast gels for electrophoresis at 140 V for 40 min, followed by blotting onto PVDF membrane under 100 V for 100 min in ice bath. Membranes were blocked with blocking buffer for 1 h at room temperature and incubated with primary antibody (1:1000 dilution) at 4 °C overnight. After washing with TBST three times (5 min per wash), the membrane was incubated with secondary antibody (1:10,000 dilution) for 1 h at room temperature. After washing with TBST six times, chemiluminescent substrate was added and incubated for 1 min. The membranes were then scanned using a blot scanner. Subsequently, the scanned membranes were stripped using Restore™ Western Blot Stripping Buffer, blocked, and incubated with the next primary antibody.

## Cell spheroids generation
Spheroids were generated using a low-adhesion U-bottom microplate, specifically the Thermo Scientific™ Nunclon™ Sphera™ 96U-well

microplate (Cat. No. 174929). KPCA-C cells (10,000 cells per 100 μL) were seeded into each well and incubated in a cell incubator for 48 h to generate spheroids.

## 3D Cell viability assay

The cytotoxicity of CyMA against the generated cell spheroids was tested using the CellTiter-Glo® 3D Cell Viability Assay, which measures ATP as an indicator of viability and generates a luminescent readout that is much more sensitive than colorimetric or fluorescence-based methods. Specifically, 100 μL of CellTiter-Glo® 3D Reagent was added into each well containing 100 μL of culture media with cell spheroids. The contents were mixed vigorously for 5 min to induce cell lysis, followed by a 25-minute incubation at room temperature to stabilize the luminescent signal. Luminescence was recorded using a microplate reader.

## Patient-derived organotypic tumor spheroids (PDOTS)

Patient tissue studies were reviewed and approved by the Institutional Review Board of Brigham and Women's Hospital. Patient-derived tumor ascites were collected by paracentesis and centrifuged to form a cell pellet. The supernatant was aspirated, and the pellet was resuspended in ACK lysis buffer (Thermo Fisher, A1049201). After maximum hemolysis was observed, PBS was added, and the process was repeated to ensure red blood cells and hemolytic products were removed. The final cell pellet was gently agitated to promote the resuspension into spheroids. The resuspended sample was filtered through a 100 μm filter. The supernatant was labeled as S2 + 3 while the filtered cell aggregates were labeled as S1, following previously established naming conventions[107]. On ice, a mixture of collagen, NaOH, phenol red, water, and PBS was made and adjusted to a pH of approximately 7.3-7.4. The spheroids were pelleted again by centrifuging at 300 g for 3 minutes. This pellet was resuspended in the collagen mixture, and 10 μL of the spheroid-collagen mixture was loaded into a microfluidic device (AIM Biotech, DAX-1) as previously described[107,108]. The devices were incubated at 37 °C for 35–40 min to allow the collagen to polymerize and form a matrix. After polymerization, 300 μL of RPMI media supplemented with 6,000 U/mL IL-2 (Miltenyi Biotec, 130097746) was divided evenly between all four ports with treatments added: 3 μM 3a, 6 μM 3a, 200 μg/mL anti-PD-1 (Fisher Scientific, 501360845) and anti-PD-L1 (Fisher Scientific, 501360846), or 3 μM of 3a with 200 μg/mL anti-PD-1 and anti-PD-L1.

After 5-7 days, live-dead analysis was performed to determine cell viability in each treatment condition. A 1:1 dilution of AO/PI stain (Nexcelcom, CS2-0106) in PBS was made, and 20–30 μL of this solution was added to each device. After 5 minutes, the devices were imaged using an inverted Nikon Eclipse Ti microscope equipped with a Nikon DS-Qi1Mc camera using NIS-Elements software. Using this software, live (acridine orange (AO)) and dead (propidium iodide (PI)) cell areas were quantified for analysis.

## Drug resistance test

KPCA-B or KPCA-C cells were divided into two culture dishes. Group 1: The KPCA cells were incubated with 3a (500 nM) for 24 hours, and the media was replaced with fresh media for 2 more days to allow the cells to expand. The cells were then sub-cultured and allowed to proliferate until they reached 80-90% confluency, followed by incubation with 3a to initiate the next cycle of cell stimulation. Group 2: KPCA cells incubated with vehicle (DMSO) instead of 3a, following the same procedure as a control. We seeded stimulated cells in from Group 1 and unstimulated KPCA cells (Group 2) in 96-well plates at $1 \times 10^5$ cells/well for 24 hours. The cells were then treated with 3a for 24 h, and the cell viability was measured using the MTT assay.

## In vivo experiments

Animal studies were conducted following the guidelines provided by the Institutional Animal Care and Use Committee (IACUC) of Brigham and Women's Hospital. Animals were maintained in a pathogen-free facility with a 12 h light/dark cycle, ambient temperature of $22 \pm 2$ °C, and humidity of $50 \pm 10\%$. Tumor engraftment was performed through intraperitoneal injection of $3.7 \times 10^6$ KPCA-B cells resuspended in a 1:1 solution of PBS and Matrigel (Corning, 354234) into 6-week C57BL/6 J female mice (Jackson Laboratories, Strain # 000664). Mice were treated via intraperitoneal injection with 3a resuspended in PBS for a final circulating blood concentration of 3 μM (0.144 mg/kg), 50 μg of Anti-PD-L1 (BioCell, BE0101) resuspended in PBS at pH 6.5 (BioCell, IP0065), 50 μg of Anti-CTLA-4 (BioCell, BE0131) resuspended in PBS at pH 7.0 (BioCell, IP0070), or a combination of 3a (0.144 mg/kg), 50 μg of Anti-PD-L1, and 50 μg of Anti-CTLA-4. Tumor burden was determined by collecting tumors through necropsy. Blood samples were obtained via cardiac puncture in EDTA coated microcentrifuge tubes (Owens & Minor, 0723365974), which were immediately centrifuged at $2000 \times g$ for 15 minutes at 4 °C. The plasma supernatant was collected and stored at $-80$ °C.

## Metabolic labeling

HeLa cells ($1.5 \times 10^5$ cells) were seeded in a confocal dish for 24 h to allow attachment. The next day, the media was replaced with fresh aliquots containing 50 μM of azido-metabolic molecule (e.g., palmitic acid, myristic acid, Ac4GlcNAc, Ac4GalNAc, Ac4ManNAc) in culture medium and add into the experimental and positive control cells. The cells were incubated for an additional 12 hours. After this, the media was replaced with fresh media containing either (CyMA + azido-metabolic molecule) or (DMSO + azido-metabolic molecule), and the cells were incubated for another 12 h. The cells were then rinsed, fixed, and a fluorophore was attached. Specifically, the AF488-alkyne was linked to the azide-modified proteins via a copper(I)-catalyzed azide-alkyne cycloaddition reaction, using a standard cell reaction buffer kit (Click-&-Go®, Catalog # CCT-1263). After washing the cells three times with PBS, they were imaged using CLSM to quantify fluorescence intensity.

## Immunoprecipitation of ubiquitinated proteins

KPCA-C cells were treated with CyMA and lysed. Ubiquitinated proteins were pulled down using Signal-Seeker ™ Ubiquitination Detection Kit, following the manufacturer's protocol. Both the input lysate and enriched ubiquitinated proteins were analyzed by immunoblotting.

## Proteomics of palmitoylated proteins

KPCA-C cells were metabolically labeled with azido-palmitic acid followed by CyMA treatment. Palmitoylated proteins were pulled down using Click-&-Go® Protein Enrichment Kit to capture azide-modified proteins according to the manufacturer's protocol. On-bead digestions were stored in 50 mM Tris buffer and sent for LC-MS/MS analysis. Three independent samples were tested for each group.

## Induction of senescent cells

HeLa cells were grown to 80% confluency and treated with 1 μg/mL of cisplatin for 24 h. After treatment, the cells were rinsed twice with PBS and cultured for an additional 6-7 days in fresh complete media, with media change every 2-3 days. The senescent HeLa cells exhibited abnormal morphology and increased SA-βGal expression level.

## ELISA

Cytokine secretion in the conditioned media was assayed using an ELISA kit, following the procedure recommended by the supplier. Opti-MEM was used as the media to avoid the external source of TGFB1 or other cytokines from FBS. TGFB1 proteins were activated according to the manufacturer's protocol before being applied to the ELISA.

## Statistics and reproducibility

All graphs were created using Origin 2021. Quantification of fluorescence intensity was conducted using Fiji. Error bars represent s.d. unless otherwise noted. For comparison between two groups, p-values

were determined using two-tailed Student's t-tests. For highly significant results where the exact value does not further impact scientific interpretation, $p < 0.0001$ is reported.

The unit of study for cell-based imaging (e.g., Figs. 2, 3, 4, and 6) is defined as individual, randomly assigned cells independently analyzed across multiple experiments. For viability, secretion, and biochemical assays (e.g., Figs. 5 and 6), the unit of study is independent wells from at least three biological replicates. In animal studies (Fig. 5n–q), the unit of study is individual mice randomly assigned to treatment groups ($n = 4$ or $n = 8$ as indicated). All representative micrographs (Figs. 2i, 3, 4, and 6) were obtained from at least three independent experiments with similar results. For all box plots presented (e.g., Fig. 2d), the center line represents the median, the box bounds indicate the 25th and 75th percentiles, and the whiskers extend to the minimum and maximum values of the dataset. All such plots are derived from n independent biological replicates as specified in the legends.

### Reporting summary

Further information on research design is available in the Nature Portfolio Reporting Summary linked to this article.

## Data availability

The data generated in this study are available in the article and Supplementary Information. The mass spectrometry-based proteomics data for palmitoyl-proteome profiling and ubiquitinated protein identification are provided in Supplementary Data 1 and 2. Source data are provided in this paper.

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

## Acknowledgements

We thank the Harvard Taplin Mass Spectrometry Facility for con-ducting LC-MS/MS analysis and the Brandeis Light Microscopy Facility (RRID:SCR_025892) for assistance. This work is partially supported by NIH CA142746 (B.X.), EY036512 (B.X.), NSF DMR-2011846 (B.X. and A.R.), Ministry of Education (Singapore) Tier 2 MOE-T2EP30221-0001 (L.L.), NIH NS103967 (A.R.) and R24 GM134210 (C.L.). We gratefully acknowledge a grant from the NIH Shared Instrumentation Program (1S10OD034395).

## Author contributions

W.T. and B.X. designed the project, synthesized the compounds, cul-tured cells, analyzed the data, and wrote the manuscript, with input from all authors. Q.Z. and Z.L. synthesized the compounds and cultured cells. K.Q. and J.D. assisted with the SIM experiments. D.M. and L.L. assisted cell experiments. T.G., D.D., and J.H. assisted the ex vivo and in vivo experiments. E.C.D and A.A.R. assisted with the culture and imaging of the Drosophila larva. N.C., D.M.L., and R.D. assisted the cell culture and analysis of cytotoxicity. C.X. and C.L. assisted with the proteomic ana-lysis. W.L., M.L., and I.A. assisted with the cell experiments. P.H. assisted the image processing.

## Competing interests

A patent application related to the Cycling Molecular Assemblies (CyMA) platform has been filed by Brandeis University (U.S. Provisional Patent Application Serial No. 63/442,344, status: pending). W.T. and B.X. are listed as inventors in this application. The remaining authors declare no competing interests.

## Additional information

**Peer review information** *Nature Communications* thanks Sudipta Basu, Yohann Boutte and the other anonymous reviewer(s) for their contribu-tion to the peer review of this work. A peer review file is available.

