## [Transparent Peer Review file · Nature Communications]

Cycling molecular assemblies for Golgi imaging and disruption

Corresponding Author: Dr Bing Xu

Version 0:

Reviewer comments:

Reviewer #1

(Remarks to the Author)

The authors described a new set of thioester peptides, derived from the previous work published in JACS in 2022 and explored their suitability for Golgi imaging and disruption. The work is a detailed study of the molecular mechanisms causing these effects. Remarkably, this work includes a palmitoylation step behind the formation of molecular assemblies. In addition, Golgi disrupters show interesting cytotoxicity and some selectivity among different cell lines, in patient-derived spheroids, and in vivo models. All in all, this is an interesting work that describes interesting features and opens up novel potential applications. However, experimental details are sometimes too concise and more detailed explanations are required in some cases.

- The synthesized compounds should be better described. There is no information about yields, amount obtained, and the NMR spectra of the compounds are lacking. The compounds are closer to small molecules than to peptides, and therefore should be fully analysed. Additionally, mass spectra should display the full window and include the m/z values on the x-axis. Moreover, the synthesis of DAN-CyMA-i or DBD-CyMA-i is not included in the manuscript.
- The LC/HRMS analysis of the peptide metabolism in cells (fig 2f) should show the chromatogram and explain how the identity was calculated (calculated mass, mass difference, etc). Is it possible to quantify the different populations, i.e. the ratio between palmitoylated/non palmitoylated/acetylated peptides? Do the authors detect other compounds, such as those derived from disulfide formation? What is displayed in Sup Fig 1k, are the chromatograms resulting from the analysis of the synthesized compounds or the traces of the compounds detected in cell lysates? What are the conditions employed for all these analyses (columns, solvents, gradients, etc)? Analogously, could it be possible to analyse the metabolism of compound 2a (Figure 3c)? Is disulfide formation preventing its palmitoylation and reducing its effects?
- The authors claim that a reversible cycling between palmitoylation and depalmitoylation may be responsible for the observed effects. However, it is not fully clear if this occurs. Thus, the high immobile fraction observed possibly suggests that thioesterases are mainly involved in the first deacetylation step, more than in a palmitoylation/depalmitoylation cycle. Once palmitoylated peptides are formed do not seem to move or diffuse out of the Golgi. Concerning these observations, it is not fully clear the effect claimed in Fig 2k, that removing the compound from the media caused fading of Golgi fluorescence. Moreover, if a cycle were involved, a thioesterase inhibitor would be expected to cause a change in peptide localization (fig 2k) for example, a shift from Golgi to cytosol? Which is the expected location of PPT1, APT1, APT2?
- Additional details are also lacking in the description of the experiments based on metabolic labelling. How was the fluorescence labeling performed? Which fluorophores and methods were employed for the coupling reaction? What exactly is measured in Fig 5b, because palmitoyl azide can also be incorporated in cellular lipids influencing the overall fluorescence observed.
- How is the rate of Golgi accumulation calculated in Figure 2e?

Other points:

The authors describe the synthesized molecules as amphiphilic peptide derivatives, although the amphiphilic character is not really clear. The hydrophilic part is rather limited. Can you please discuss it?

Can the authors explain and expand the discussion about the claimed heightened sensitivity of Ras-driven cancer cells mentioned in Figure 5h?

Reviewer #2

(Remarks to the Author)

Golgi apparatus has emerged as one of the most interesting organelles in the progression and development of cancer. Hence imaging and targeting Golgi will be an interesting strategy for cancer therapeutics. However, selective targeting of Golgi remained a major challenge. Herein, the authors report cycling molecular assemblies (CyMA) for rapid Golgi imaging and cell-selective Golgi disruption in different cancer cells. Authors demonstrated that dynamic supramolecule assemblies provide an active and selective strategy for Golgi-targeting by interfering with Golgi functions. This approach may be applicable to targeting other organelles by utilizing alternative enzyme switches. This is a very comprehensive work with sound experimental support. Minor revision suggested as follows:

The authors should provide full chemical characterization of all the new molecules and peptides synthesized in this manuscript including ¹H, ¹³C NMR data and spectra along with the HR-MS data. The data should be incorporated into the supporting information.

Reviewer #3

(Remarks to the Author)

The manuscript by Tan et al. describes the use of cycling molecular assemblies (CyMA) to rapidly label the Golgi apparatus without the need for fluorescent protein fusions or reporters. This is achieved by incubating cells with thiopeptides that undergo palmitoylation at the Golgi. The authors also demonstrate that this strategy can be used to disrupt Golgi function and induce cell death by replacing the fluorophore with a biphenyl motif.

Overall, I am impressed by the quantity and quality of the data presented in this paper. The illustrations and figures are well designed, attractive and convincing, and everything is quantified and statistically tested. This makes it a strong candidate for publication in a journal such as Nature. Comm. I think these tools will be very useful for the Golgi research community.

However, I have a couple of questions and requests for clarification that I would like to put to the authors.

1- The NBD version is fluorescent and localized to the Golgi, whereas the NTG version is not fluorescent. How can you be sure that this version is localized to the Golgi as well? I have a couple of questions related to this later on.

2- If the probe is trafficked to the Golgi independently of endocytosis, why is the uptake of the probe higher in the vps35 mutant, which has an endosome-to-Golgi trafficking defect, rather than being equal? More generally, I think it would be worthwhile looking at mutants of clathrin- and dynamin-related endocytosis.

3- Many approaches rely on drug treatment, which I don't consider problematic. However, as suggested in my previous point on endocytosis, some conclusions may need to be supported by genetic approaches, as well as for ER-Golgi trafficking. Your data suggest that Golgi disruption occurs by mechanisms other than ER-Golgi redistribution, and indicate disruption to anterograde and retrograde transport. However, it would be good to have some genetic confirmation of this.

4- Finally, my main point is on S-palmitoylation, which brings me back to my first point. N-palmitoylation is considered irreversible, whereas S-palmitoylation is reversible. There is therefore a discrepancy between this and the phenotypes observed with the NTG version of the probe. This probe, which disrupts the Golgi, has a strong effect on the cell, even leading to cell death. The authors demonstrate that palmitoylation is responsible for the probe's inability to diffuse at the Golgi. At the same time, I would expect a less drastic effect on the cell if palmitoylation were reversible. Another point is that the authors observed a general reduction in palmitoylation with the NTG version of the probe, which is palmitoylated itself. Does the probe detach from the Golgi at some point? What is the lifetime of the NTG probe at the Golgi? The data on RAS, AKT and mTOR at the end of the paper suggest that cell death and proliferation defects are due to a general decrease in palmitoylation, but are all these defects linked to the Golgi? DHHCs localize to different membranes within the cell: the ER (DHHC1 and 6), the ER-Golgi (DHHC2 and 9), the Golgi (DHHC7 and 8), and even the plasma membrane (DHHC5, 20, and 21). How can you therefore be completely sure that the effects are linked to the Golgi only? You cannot see the NTG probe, so it could localize to other membranes besides the Golgi. Palmitoylation occurs not only at the Golgi, and the phenotypic effects could be due to processes other than those related to the Golgi.

Overall, I really liked this paper, which I think is supported by strong evidence and quantification. However, I would encourage the authors to clarify the final point that I raised, otherwise this probe will be used in future papers to disrupt the function of the Golgi apparatus and draw conclusions from this. However, this could alter much more than the Golgi apparatus, which could lead to inappropriate statements being made.

Version 1:

Reviewer comments:

Reviewer #1

(Remarks to the Author)

As I already mentioned in my first review, overall, this is a good work, with a substantial amount of interesting data. The reported results may be of interest to the scientific community, and therefore, it may deserve publication. However, some of

the points I raised in my previous review have not been clarified yet.

The authors have included the structural characterization for compounds 1a and 3a, but the ones corresponding to the other synthesized compounds (2b, 2c, 2d, 4a-4c, 5 6, DAN-CyMA-i and DBD-CyMA) are still lacking. A correct characterization of synthesized molecules is always required, and this is even more important if their biological effect is studied.

Moreover, for the identification of palmitoylated Peptide-SPalm resulting from different CyMA (Figure S1k), can you be sure that the palmitoylated peptides will be detectable under the used conditions (C18 column)? Do you have synthesized samples to analyse them and compare their retention times with the ones observed in the cell lysates? The unambiguous detection of the formed palmitoylated peptides is crucial for the project, and it may be challenging in a sample derived from a cell lysate if the required reference samples are lacking.

Regarding the discussion about the inability of the probe to diffuse at the Golgi and the dynamic palmitoylation cycle, pointed out in my first revision and also commented on by Reviewer 3, I am not convinced by the explanation provided. A dynamic reversible S-palmitoylation associated with a high immobile fraction would require that the thioesterases are also located in the Golgi, which does seem to be the case (the main effect is observed with PPT1, which is mainly a lysosomal enzyme). All in all, palmitoylation and Golgi localization may be clear, but not its reversibility. This point is unclear, and the discussion sounds too speculative at the moment, and therefore it should be revised.

Reviewer #2

(Remarks to the Author)

The authors answered all the concerns raised by the reviewer. This manuscript can be now considered for publication in this current form.

Reviewer #3

(Remarks to the Author)

The authors have now addressed my concerns to my satisfaction, either through experimental approaches, by revising the text, or by exercising more caution in their conclusions and clarifying the limitations of their work. I therefore have no further concerns.

Version 2:

Reviewer comments:

Reviewer #1

(Remarks to the Author)

The authors have addressed my concerns satisfactorily, and the manuscript has been revised accordingly. I recommend the acceptance of the manuscript.

The following are our responses to the comments (in *Italics*) of the reviewer and the changes (underlined) in the manuscripts.

A) Reviewer 1:

[The authors described a new set of thioester peptides, derived from the previous work published in JACS in 2022 and explored their suitability for Golgi imaging and disruption. The work is a detailed study of the molecular mechanisms causing these effects. Remarkably, this work includes a palmitoylation step behind the formation of molecular assemblies. In addition, Golgi disrupters show interesting cytotoxicity and some selectivity among different cell lines, in patient-derived spheroids, and in vivo models. All in all, this is an interesting work that describes interesting features and opens up novel potential applications. However, experimental details are sometimes too concise and more detailed explanations are required in some cases.]

We appreciate the insightful and constructive comments from the reviewer. We agree with the reviewer that the study describes interesting features and opens up novel potential applications. We have revised the manuscript to enhance clarity and provide greater depth in both the main text and the supplementary methods.

[• The synthesized compounds should be better described. There is no information about yields, amount obtained, and the NMR spectra of the compounds are lacking. The compounds are closer to small molecules than to peptides, and therefore should be fully analysed.]

We thank the reviewer for the comments. We used standard Fmoc solid-phase peptide synthesis and have now clarified this in the revised methods. While overall yields vary due to multi-step coupling and cleavage processes, we emphasize purity and identity for biological evaluation, and we now provide representative NMR data as requested.

Addition in SI:

Supplementary Figure S15. ¹H NMR spectrum of **1a** in DMSO-*d*₆. δ 9.30 (brs, 1H), 8.56 (d, *J* = 7.4 Hz, 1H), 8.47 (d, *J* = 8.9 Hz, 1H), 8.26 (d, *J* = 8.7 Hz, 1H), 7.31 – 7.16

(m, 7H), 7.13 (t, $J = 7.3$ Hz, 2H), 7.10 – 7.03 (m, 1H), 6.32 (d, $J = 9.0$ Hz, 1H), 4.59 (td, $J = 9.8, 4.2$ Hz, 1H), 4.46 (td, $J = 8.2, 6.2$ Hz, 1H), 4.08 (t, $J = 6.4$ Hz, 2H), 3.52 (brs, 2H), 3.08 – 2.90 (m, 5H), 2.68 (dd, $J = 13.9, 10.1$ Hz, 1H), 2.53 – 2.45 (m, 2H), 2.32 (s, 3H).

Supplementary Figure S16. APT ^{13}C NMR spectrum of **1a** in $\text{DMSO-}d_6$. δ 194.8, 171.4, 171.0, 169.5, 144.82, 144.75, 144.3, 144.1, 137.8, 137.7, 137.0, 129.10 (2C), 129.07 (2C), 128.3 (2C), 127.8 (2C), 126.6, 126.1, 99.3, 62.8, 53.6, 53.4, 39.8, 37.7, 36.5, 33.7, 30.5, 27.2.

Supplementary Figure S17. ¹H NMR spectrum of **3a** in DMSO-*d*₆. δ 8.60 (d, J = 8.5 Hz, 1H), 8.56 (d, J = 7.5 Hz, 1H), 7.88 (d, J = 8.4 Hz, 2H), 7.78 – 7.69 (m, 4H), 7.53 – 7.45 (m, 2H), 7.45 – 7.37 (m, 1H), 7.41 – 7.32 (m, 2H), 7.28 – 7.23 (m, 6H), 7.23 – 7.19 (m, 1H), 7.18 – 7.13 (m, 1H), 4.78 (td, J = 10.0, 3.9 Hz, 1H), 4.52 (td, J = 8.0, 6.1 Hz, 1H), 4.10 (t, J = 6.4 Hz, 2H), 3.11 (dd, J = 6.7, 3.0 Hz, 1H), 3.11 – 3.04 (m, 1H), 3.07 – 2.98 (m, 3H), 3.01 – 2.93 (m, 1H), 2.32 (s, 3H).

Supplementary Figure S18. APT ¹³C NMR spectrum of **3a** in DMSO-*d*₆. δ 194.8, 171.7, 171.0, 165.8, 142.8, 139.1, 138.4, 137.0, 132.8, 129.1(2C), 129.1(2C), 129.0(2C), 128.3(2C), 128.1(2C), 128.0(two peaks overlapped, 3C), 126.8(2C), 126.6, 126.4(2C), 126.2, 62.8, 54.5, 53.7, 37.0, 36.5, 30.4, 27.2.

Addition in SI:

Proton nuclear magnetic resonance (¹H NMR) spectra and Attached Proton Test (APT) carbon nuclear magnetic resonance (¹³C NMR) spectra were recorded on Advance NEO 400 (400 MHz). Chemical shifts for protons are reported in parts per million downfield from tetramethylsilane and are referenced to the NMR solvent residual peak (DMSO-*d*₆ δ 2.50). Chemical shifts for carbons are reported in parts per million downfield from tetramethylsilane and are referenced to the carbon resonances of the NMR solvent (DMSO-*d*₆ δ 39.5).

Addition:

We gratefully acknowledge a grant from the NIH Shared Instrumentation Program (1S10OD034395).

[Additionally, mass spectra should display the full window and include the m/z values on the x-axis. Moreover, the synthesis of DAN-CyMA-i or DBD-CyMA-i is not included in the manuscript.]

We thank the reviewer for the comments. We agree with the reviewer's suggestions. We have updated the LC-HRMS spectra in the Supplementary Information to include the m/z axis and have now added the full synthetic details for DAN-CyMA-i and DBD-CyMA-i in the Methods section. We appreciate the reviewer for pointing out these opportunities for improvement.

Original in SI:

Supplementary Figure S13. LC-HRMS of the synthesized CyMA-i, 1a, 2a-2d, 5.

Supplementary Figure S14. LC-HRMS of the synthesized CyMA-d, **3a**, **4a-4c**, **6**.

Revision in SI:

Supplementary Figure S13. LC-HRMS of the synthesized CyMA-i, **1a**, **2a-2d**, **5**, **DAN-CyMA-i**, **DBD-CyMA-i**.

Supplementary Figure S14. LC-HRMS of the synthesized CyMA-d, 3a, 4a-4c, 6.

Original in SI:

Synthesis of NBD- β -Alanine

Scheme S1. Synthetic procedure of NBD- β -alanine.

To a 10 mL aqueous solution of β -alanine (5.5 mmol, 490 mg) and potassium carbonate (16.5 mmol, 2.07g), NBD-Cl (5 mmol, 1g) in 60 mL of MeOH was added dropwise with stirring under nitrogen gas protection. After stirring at room temperature for 6 hours, the methanol was removed by a rotary evaporator and the residual solution was acidified to pH 3 using 1N HCl. The acidic aqueous solution was then extracted by diethyl ether. The combined organic solution was dried over anhydrous sodium sulfate and concentrated by rotary

evaporator. The resulting dark-yellow powder (NBD- β -alanine) was used directly for solid phase peptide synthesis.

Revision in SI:

Synthesis of NBD- β -Alanine, DAN- β -Alanine, and DBD- β -Alanine

Scheme S1. Synthetic procedure of NBD- β -alanine, DAN- β -alanine, and DBD- β -alanine.

To a 10 mL aqueous solution of β -alanine (5.5 mmol, 490 mg) and potassium carbonate (16.5 mmol, 2.07g), NBD-Cl (5 mmol, 1g) or dansyl chloride (DAN-Cl) (5 mmol, 1.35 g) or 4-(N,N-dimethylaminosulfonyl)-7-fluoro-2,1,3-benzoxadiazole (DBD-F) (5 mmol, 1.30 g) in 60 mL of MeOH was added dropwise with stirring under nitrogen gas protection. After stirring at room temperature for 6 hours, the methanol was removed by a rotary evaporator and the residual solution was acidified to pH 3 using 1N HCl. The acidic aqueous solution was then extracted by diethyl ether. The combined organic solution was dried over anhydrous sodium sulfate and concentrated by rotary evaporator. The resulting dark-yellow powder (NBD- β -alanine) or pale-yellow powder (DAN- β -alanine) or yellow powder (DBD- β -alanine) was used directly for solid phase peptide synthesis.

Addition in SI:

Synthesis of DAN-CyMA-i, DBD-CyMA-i

We used standard Fmoc chemistry for solid phase peptide synthesis (SPPS) with 2-chlorotrityl chloride resin and Fmoc-protected amino acids with appropriately protected side chains. Briefly, the 2-Cl resin (1 g) was swelling in dry DCM for 30 minutes, followed by loading the first amino acid onto the resin. Next, 10 equivalents of 2-mercaptoethanol dissolved in DCM/DMF (v:v=1:1) were added

into the SPPS reactor and incubated with the resin overnight at room temperature. The thiol group of 2-mercaptoethanol is selectively attached to the resin due to its high nucleophilicity. After discarding the liquid phase, the resin with mercaptoethanol linker was washed five times with DMF. Subsequently, Fmoc-protected D-phenylalanine (1.5 equiv.), along with N,N'-dicyclohexylcarbodiimide (DCC, 2 equiv.) and a catalytic amount of 4-Dimethylaminopyridine (DMAP) dissolved in DMF, was added to the reactor and incubated with the resin overnight at room temperature. The following day, the Fmoc group was removed with 20% piperidine in DMF, and the next Fmoc-protected D-phenylalanine was coupled to the free amino group using HBTU as the coupling reagent. After the removal of Fmoc-protecting group, the N-terminus of the peptide was capped with either DAN- β -alanine or DBD- β -alanine on the resin. The peptide chain was cleaved from the resin by 95% TFA (95% TFA, 2.5% TIPS, 2.5% H₂O) for 1 hour, followed by chilling in iced water bath and flushed with nitrogen. Then, 4 equivalents of acetyl chloride were added dropwise into the TFA solution with the peptide chain, and the temperature was raised to room temperature. The reaction was left standing for 3 hours and then quenched with ice-cold water. After the solvent was removed by rotary evaporation, the resulting oily product was purified directly by RP-HPLC to give DAN-CyMA-i or DBD-CyMA-i with thioester bonds.

[• *The LC/HRMS analysis of the peptide metabolism in cells (fig 2f) should show the chromatogram and explain how the identity was calculated (calculated mass, mass difference, etc).*]

We thank the reviewer for the comments. In the original submission, Supplementary Figure S1k was the extracted-ion chromatograms used to illustrate the different retention times of the key metabolites. To address the reviewer's comment, we have now updated the caption for the Supplementary Figure S1k.

Original:

Supplementary Figure S1k. Retention time comparison of the corresponding Peptide-SH and Peptide-SPalm resulting from different CyMA (**1a**, **2a**, and **3a**) using LC-HRMS.

Revision:

Supplementary Figure S1k. Representative extracted-ion chromatograms from the LC-HRMS analysis of HeLa cell lysates, showing the retention time comparison of the corresponding Peptide-SH and Peptide-SPalm resulting from different CyMA (**1a**, **2a**, and **3a**).

To clarify how the identity of the metabolites was confirmed, we have also added a detailed description of our analytical method to the revised Supplementary Information.

As is now explained, the identity of each compound was confirmed by matching the experimentally observed high-resolution m/z with the calculated theoretical exact mass. We further verified these assignments by confirming that the observed isotopic distribution pattern of the peaks in the mass spectrum matched the theoretical pattern predicted for the compound's elemental formula. We hope these additions provide the requested detail and strengthen the manuscript.

Addition in SI:

The identity of each compound was confirmed by matching the experimentally observed high-resolution mass-to-charge ratio (m/z) with the calculated theoretical exact mass. We further verified these assignments by confirming that the observed isotopic distribution pattern of the peaks in the mass spectrum matched the theoretical pattern predicted for the compound's elemental formula.

[Is it possible to quantify the different populations, i.e. the ratio between palmitoylated/non palmitoylated/acetylated peptides?]

We thank the reviewer for the comments. We agree that this quantitative analysis is a critical next step for our research, and we have now added a sentence to the Discussion section to highlight this as a key future direction. We thank the reviewer for this valuable suggestion for our future studies.

Original:

Looking forward, the concept of establishing enzyme-controlled, futile assembly-disassembly cycles at defined subcellular locations and consuming endogenous metabolites provides a broadly applicable framework. Characterizing the precise nature of the observed subcellular reorganizations and their cargo will be a critical next step.

Revision:

Looking forward, the concept of establishing enzyme-controlled, futile assembly-disassembly cycles at defined subcellular locations and consuming endogenous metabolites provides a broadly applicable framework. To realize the full potential of this approach, several critical next steps are required. These include characterizing the precise nature of the observed subcellular reorganizations and their cargo, as well as undertaking a detailed quantitative analysis of this dynamic system. Furthermore, a detailed investigation into the various upstream endocytic pathways and specific proteins most affected by CyMA-d using genetic approaches would also be a valuable future direction.

[Do the authors detect other compounds, such as those derived from disulfide formation?]

We thank the reviewer for the comments. In our LC-HRMS analysis of the cell lysates, we found no evidence of alternative S-lipidation (e.g., S-farnesylation or S-geranylgeranylation) or disulfide-linked dimers. Under our experimental conditions (1

μM precursor, 30-minute treatment), rapid enzyme-driven S-palmitoylation appears to dominate, which likely minimizes disulfide formation. We have clarified in the revised manuscript how this differs from our previous findings (JACS, 2022), to avoid potential confusion.

Original:

To obtain direct evidence of thiopeptide derivative palmitoylation, we analyzed the lysates of HeLa cells treated with **1a**. Liquid chromatography coupled to high-resolution mass spectrometry (LC-HRMS) confirmed CyMA lipidation at the Golgi by identifying peaks for both the deacylated product (Peptide-SH) and the palmitoylated CyMA (Peptide-SPalm) of **1a** (Fig. 2f, Supplementary Fig. S1k).

Revision:

To obtain direct evidence of thiopeptide derivative palmitoylation, we analyzed the lysates of HeLa cells treated with **1a**. Liquid chromatography coupled to high-resolution mass spectrometry (LC-HRMS) confirmed CyMA lipidation at the Golgi by identifying peaks for both the deacylated product (Peptide-SH) and the palmitoylated CyMA (Peptide-SPalm) of **1a** (Fig. 2f, Supplementary Fig. S1k). Under these mild conditions (1 μM for 30 minutes), we did not detect other potential products, such as those from alternative S-lipidation or disulfide formation, suggesting that the rapid, enzyme-driven S-palmitoylation is the dominant pathway for the free thiol intermediate.

[What is displayed in Sup Fig 1k, are the chromatograms resulting from the analysis of the synthesized compounds or the traces of the compounds detected in cell lysates? What are the conditions employed for all these analyses (columns, solvents, gradients, etc)?]

We thank the reviewer for the comments. Supplementary Figure S1k shows the extracted-ion chromatogram from the analysis of the cell lysate. We have now revised the caption to enhance the clarity of the data (is from the cell lysate analysis).

We also revised the Supplementary Information to include the full experimental details of the LC-HRMS analysis, including the column, solvents, and gradient conditions used. We appreciate the reviewer helping us improve the completeness of our methods section.

Original in SI:

Supplementary Figure S1k. Retention time comparison of the corresponding Peptide-SH and Peptide-SPalm resulting from different CyMA (**1a**, **2a**, and **3a**) using LC-HRMS.

Revision in SI:

Supplementary Figure S1k. Representative extracted-ion chromatograms from the LC-HRMS analysis of HeLa cell lysates, showing the retention time

comparison of the corresponding Peptide-SH and Peptide-SPalm resulting from different CyMA (**1a**, **2a**, and **3a**).

Original in SI:

A total of 1.2×10^7 cells were treated with 1 μ M of CyMA or vehicle (DMSO) for 30 minutes...and the remaining solid was dissolved in 150 μ L of methanol and sent for LC/HR-MS analysis.

Revision in SI:

A total of 1.2×10^7 cells were treated with 1 μ M of CyMA or vehicle (DMSO) for 30 minutes...and the remaining solid was dissolved in 150 μ L of methanol and sent for LC/HR-MS analysis. The analysis was performed on a C18 reverse phase column using water (0.1% formic acid) and acetonitrile (0.1% formic acid) as the mobile phases, with a gradient of 1% to 99% acetonitrile over 14 minutes. The identity of each compound was confirmed by matching the experimentally observed high-resolution mass-to-charge ratio (m/z) with the calculated theoretical exact mass. We further verified these assignments by confirming that the observed isotopic distribution pattern of the peaks in the mass spectrum matched the theoretical pattern predicted for the compound's elemental formula.

[Analogously, could it be possible to analyse the metabolism of compound 2a (Figure 3c)? Is disulfide formation preventing its palmitoylation and reducing its effects?]

We thank the reviewer for the comments. We appreciate this thoughtful suggestion. We have now clarified in the text that metabolism data for **2a** were included (Supplementary Fig. S1k) and explicitly discuss potential disulfide formation as a contributing factor.

Original:

Importantly, **2a** produced significantly less fluorescence at Golgi than **1a** under identical conditions. Even after 30 minutes of treatment with 500 nM **2a**, only weak perinuclear fluorescence was observed (Fig. 3d). This suggests that free thiols, prior to reaching the Golgi, may engage in additional reactions beyond those that lead to Golgi localization, and that in-situ thiol formation and reversible S-palmitoylation—as in the case of **1a**—are key to achieving high targeting efficiency.

Revision:

Importantly, **2a** produced significantly less fluorescence at Golgi than **1a** under identical conditions. Even after 30 minutes of treatment with 500 nM **2a**, only weak perinuclear fluorescence was observed (Fig. 3d). Although our LC-HRMS analysis confirmed that **2a** is a substrate for palmitoylation (Supplementary Fig. S1k), its lower efficacy suggests that when administered exogenously, free thiols, prior to reaching the Golgi, may engage in additional reactions beyond those that

lead to Golgi localization, such as the formation of disulfide bonds with proteins or being oxidized. This suggests that in-situ thiol formation and reversible S-palmitoylation—as in the case of **1a**—are key to achieving high targeting efficiency.

[• The authors claim that a reversible cycling between palmitoylation and depalmitoylation may be responsible for the observed effects. However, it is not fully clear if this occurs. Thus, the high immobile fraction observed possibly suggests that thioesterases are mainly involved in the first deacetylation step, more than in a palmitoylation/depalmitoylation cycle. Once palmitoylated peptides are formed do not seem to move or diffuse out of the Golgi. Concerning these observations, it is not fully clear the effect claimed in Fig 2k, that removing the compound from the media caused fading of Golgi fluorescence. Moreover, if a cycle were involved, a thioesterase inhibitor would be expected to cause a change in peptide localization (fig 2k) for example, a shift from Golgi to cytosol? Which is the expected location of PPT1, APT1, APT2?]

We thank the reviewer for highlighting this important point. We have expanded the text to clarify that the apparent immobile fraction reflects the stability of the macroscopic assemblies, whereas individual molecules at their surface undergo dynamic turnover via reversible palmitoylation. This explanation has been added to the revised manuscript.

Original:

CyMA is non-diffusive. To assess the diffusion of CyMA, we conducted fluorescence recovery after photobleaching (FRAP) on CyMA localized at the Golgi. A concentration of 2 μM of **1a** was selected, as it does not induce Golgi fragmentation under these conditions. The results showed that the photobleached region remained dark after 60 seconds. The mobile fraction (M_f) was 0.08, and the immobile fraction (IM_f) was 0.92, indicating CyMA are solid-like and hardly diffuse from the Golgi. The time-dependent increase of IM_f correlated with CyMA accumulation. Control FRAP with C6-NBD-ceramide showed faster fluorescence recovery and yielded a lower IM_f of 0.62 (Fig. 2l), suggesting that CyMA, rather than Golgi compartmental stability, account for the high IM_f .

Revision:

CyMA is non-diffusive. To assess the diffusion of CyMA, we conducted fluorescence recovery after photobleaching (FRAP) on CyMA localized at the Golgi. A concentration of 2 μM of **1a** was selected, as it does not induce Golgi fragmentation under these conditions. The results showed that the photobleached region remained dark after 60 seconds. The mobile fraction (M_f) was 0.08, and the immobile fraction (IM_f) was 0.92, indicating CyMA are solid-like and hardly diffuse from the Golgi. The time-dependent increase of IM_f correlated with CyMA accumulation. Control FRAP with C6-NBD-ceramide showed faster fluorescence recovery and yielded a lower IM_f of 0.62 (Fig. 2l),

suggesting that CyMA, rather than Golgi compartmental stability, account for the high IM. The high immobile fraction, indicative of a solid-like state, can be reconciled with the dynamic, reversible S-palmitoylation cycle by distinguishing the behavior of individual molecules from the bulk assembly. The dynamic cycling represents the constant turnover of individual molecules at the interface of the assembly, while the high immobile fraction reflects the stability of the large, macroscopic core that is kinetically trapped at the Golgi. This model also explains the assembly's net disassembly upon precursor removal (Fig. 2k), which disrupts the system's steady state.

[• Additional details are also lacking in the description of the experiments based on metabolic labelling. How was the fluorescence labeling performed? Which fluorophores and methods were employed for the coupling reaction?]

We thank the reviewer for the comments. The details for the fluorescence labeling, including the use of an AF488-alkyne fluorophore and a standard click reaction method, were included in the "Materials" and "Metabolic labeling" sections of the original Supplementary Information. However, to make this information more explicit and consolidated, we have now expanded the "Metabolic labeling" section to provide an even more detailed description of the procedure.

Original in SI:

The cells were then rinsed, fixed, and fluorophore (AF488-alkyne) was linked to the azido incorporated proteins using a standard click reaction. After washing the cells three times with PBS, they were imaged using CLSM to quantify fluorescence intensity.

Revision in SI:

The cells were then rinsed, fixed, and the fluorophore was attached. Specifically, the AF488-alkyne was linked to the azide-modified proteins via a copper(I)-catalyzed azide-alkyne cycloaddition reaction, using a standard cell reaction buffer kit (Click-&-Go®, Catalog # CCT-1263). After washing the cells three times with PBS, they were imaged using CLSM to quantify fluorescence intensity.

[What exactly is measured in Fig 5b, because palmitoyl azide can also be incorporated in cellular lipids influencing the overall fluorescence observed.]

The reviewer raises an excellent point about the specificity of the palmitoyl-azide probe. We agree that this probe can also be incorporated into cellular lipids, which could influence the overall fluorescence observed in Figure 5b. However, this metabolic labeling strategy is a widely established and accepted method for studying the global palmitoyl-proteome, and it has been used in numerous foundational papers in the field (e.g., 10.1038/nchembio.1392, 10.1038/nmeth.1293).

More importantly, to provide robust and specific evidence supporting the fluorescence data, we also performed a palmitoyl-proteome analysis using LC-MS/MS (shown in Figure 5d). This proteomic data serves as a solid, orthogonal validation, confirming that protein S-palmitoylation is indeed globally reduced. We have now revised the manuscript to include this discussion and the appropriate references, clarifying that the fluorescence data is supported by our more specific mass spectrometry results. We hope these revisions have strengthened the manuscript.

Original:

We investigated these effects using metabolic labeling (Fig. 5a)⁵⁵⁻⁵⁷ and analyzed the outcomes through CLSM and LC-MS/MS. We observed a concentration-dependent reduction in palmitoylated proteins, indicating that CyMA-d impair lipidation (Fig. 5b). Glycosylation analysis revealed a global decrease in O-GlcNAcylation, while total sialylation levels appeared unaffected (Fig. 5b). We next performed palmitoyl-proteome profiling in KPCA-C cells treated with **3a** using LC-MS/MS (Supplementary table S1), which confirmed global reductions in palmitoylation (Fig. 5d).

Revision:

We investigated these effects using metabolic labeling (Fig. 5a)⁵⁵⁻⁵⁷ and analyzed the outcomes through CLSM and LC-MS/MS. We observed a concentration-dependent reduction in fluorescence from the incorporated palmitoyl-azide probe, suggesting that CyMA-d impair lipidation (Fig. 5b). While this metabolic labeling strategy is a widely established method⁵⁸⁻⁶⁰, we acknowledge that the probe can also be incorporated into cellular lipids. To provide more specific and robust evidence for the effect on protein lipidation, we therefore performed palmitoyl-proteome profiling in KPCA-C cells treated with **3a** using LC-MS/MS (Supplementary table S1), which confirmed global reductions in palmitoylation (Fig. 5d). Glycosylation analysis also revealed a global decrease in O-GlcNAcylation, while total sialylation levels appeared unaffected (Fig. 5b).

Original:

We next performed palmitoyl-proteome profiling in KPCA-C cells treated with **3a** using LC-MS/MS (Supplementary table S1), which confirmed global reductions in palmitoylation (Fig. 5d). Notably, several Ras isoforms (HRas, NRas, RRas), as key oncogenic GTPases, were among the affected proteins, their oncogenic functions likely being impaired as a consequence of defective lipidation.

Revision:

A detailed analysis of LC-MS/MS results (Fig. 5d) reveals several Ras isoforms (HRas, NRas, RRas), as key oncogenic GTPases, were among the affected

proteins, their oncogenic functions likely being impaired as a consequence of defective lipidation.

Addition:

58. Gao, X. & Hannoush, R. N. Single-cell imaging of Wnt palmitoylation by the acyltransferase porcupine. Nat. Chem. Biol. 10, 61-68 (2014).

59. Martin, B. R., Wang, C., Adibekian, A., Tully, S. E. & Cravatt, B. F. Global profiling of dynamic protein palmitoylation. Nat. Methods 9, 84-89 (2012).

60. Kimura, T. et al. Subcellular Analysis of Fatty Acid Metabolism Using Organelle-Selective Click Chemistry. J. Am. Chem. Soc 147, 22284-22289 (2025).

[How is the rate of Golgi accumulation calculated in Figure 2e?]

We thank the reviewer for the comments. The rate of accumulation is represented by the slope of the curves in Figure 2e, which plot the mean fluorescence intensity at the Golgi over time. This data was calculated using a custom image analysis pipeline, as detailed in the "Quantification of fluorescence intensity at Golgi" section of the Supplementary Information.

“Image processing was conducted to extract single-cell responses. The image pixel values are scaled to be between [0, 1] with 1 indicates the original intensity of 255. ... Using the Golgi mask, reaction signals were classified as occurring inside or outside the Golgi.”

The final curves in Figure 2e show the average of these intensity measurements over time from multiple cells, allowing for a direct comparison of the accumulation kinetics between different conditions.

[Other points:

The authors describe the synthesized molecules as amphiphilic peptide derivatives, although the amphiphilic character is not really clear. The hydrophilic part is rather limited. Can you please discuss it?]

We thank the reviewer for the comments. The reviewer is correct that the precursor molecule (**1a**) has a limited hydrophilic part. To address the concern of the reviewer, we add a sentence to discuss the amphiphilicity of the peptide.

Addition:

Although the precursor molecule (**1a**) has a limited hydrophilic part, the polar peptide backbone renders it amphiphilic and facilitates cell entry.

[Can the authors explain and expand the discussion about the claimed heightened sensitivity of Ras-driven cancer cells mentioned in Figure 5h?]

We thank the reviewer for the comments. The heightened sensitivity of RAS-driven cancer cells likely reflects the fundamental dependence of these oncoproteins on post-translational lipidation. Proper membrane localization and signaling of RAS isoforms require S-palmitoylation at the Golgi apparatus. As shown in Figure 5, CyMA-d

treatment disrupts Golgi integrity and impairs global S-palmitoylation. We hypothesize that this interferes with the processing and membrane anchoring of RAS, thereby collapsing downstream survival pathways in cancer cells addicted to oncogenic RAS signaling. We have expanded the Discussion to clarify this point.

Original:

This pleotropic action is particularly effective in complex disease states, as evidenced by the heightened sensitivity of RAS-driven cancer cells to CyMA-d (Fig. 5h).

Revision:

This pleotropic action is particularly effective in complex disease states, as evidenced by the heightened sensitivity of RAS-driven cancer cells to CyMA-d (Fig. 5h). Such sensitivity likely reflects the fundamental dependence of RAS oncoproteins on post-translational lipidation for their function^{43,93}. Proper membrane localization and signaling of RAS isoforms require S-palmitoylation at the Golgi; therefore, the disruption of Golgi integrity and the impairment of global S-palmitoylation by CyMA-d likely interfere with RAS processing and membrane anchoring, leading to the collapse of downstream survival pathways in these RAS-addicted cancer cells.

Addition:

93. Goodwin, J. S. et al. Depalmitoylated Ras traffics to and from the Golgi complex via a nonvesicular pathway. J. Cell Biol. 170, 261-272 (2005).

B) Reviewer 2:

[Golgi apparatus has emerged as one of the most interesting organelles in the progression and development of cancer. Hence imaging and targeting Golgi will be an interesting strategy for cancer therapeutics. However, selective targeting of Golgi remained a major challenge. Herein, the authors report cycling molecular assemblies (CyMA) for rapid Golgi imaging and cell-selective Golgi disruption in different cancer cells. Authors demonstrated that dynamic supramolecule assemblies provide an active and selective strategy for Golgi-targeting by interfering with Golgi functions. This approach may be applicable to targeting other organelles by utilizing alternative enzyme switches. This is a very comprehensive work with sound experimental support. Minor revision suggested as follows:]

We appreciate the insightful and constructive comments from the reviewer. We agree with the reviewer that this work may be applicable to targeting other organelles by utilizing alternative enzyme switches.

[The authors should provide full chemical characterization of all the new molecules and peptides synthesized in this manuscript including 1H, 13C NMR data and spectra along with the HR-MS data. The data should be incorporated into the supporting information.]

We thank the reviewer for the comments. Following the suggestion of the reviewer, we added the ^1H and ^{13}C NMR of the two key compounds (See reply to Reviewer 1).

C) Reviewer 3:

[The manuscript by Tan et al. describes the use of cycling molecular assemblies (CyMA) to rapidly label the Golgi apparatus without the need for fluorescent protein fusions or reporters. This is achieved by incubating cells with thiopeptides that undergo palmitoylation at the Golgi. The authors also demonstrate that this strategy can be used to disrupt Golgi function and induce cell death by replacing the fluorophore with a biphenyl motif.

Overall, I am impressed by the quantity and quality of the data presented in this paper. The illustrations and figures are well designed, attractive and convincing, and everything is quantified and statistically tested. This makes it a strong candidate for publication in a journal such as Nature. Comm. I think these tools will be very useful for the Golgi research community. However, I have a couple of questions and requests for clarification that I would like to put to the authors.]

We appreciate the insightful and constructive comments from the reviewer. We agree with the reviewer that this work will be useful for the Golgi research community. We included more detailed studies and explanations to strengthen our manuscript.

[1- The NBD version is fluorescent and localized to the Golgi, whereas the NTG version is not fluorescent. How can you be sure that this version is localized to the Golgi as well? I have a couple of questions related to this later on.]

We thank the reviewer for the comments. We agree with the reviewer that it is essential to establish that the non-fluorescent therapeutic compound (**3a**) localizes to the Golgi, similar to its fluorescent imaging counterpart (**1a**). We have two lines of evidence to support this conclusion.

First, to visualize its localization, we synthesized a fluorescent analogue of **3a** where one benzene ring of the phenylalanine residue was replaced with an NBD fluorophore, keeping the structure otherwise identical. As we have now shown in Supplementary Figure S4b (originally Supplementary Figure S4c), this fluorescent analogue (**3a'**) shows clear and rapid accumulation in the Golgi.

Second, we confirmed that compound **3a** is processed by the same Golgi-resident enzymatic machinery that is required for targeting. Our LC-HRMS analysis of cell lysates (Supplementary Figure S4c, originally Supplementary Figure S4b) shows that **3a** is successfully palmitoylated in cells, which provides another biochemical evidence that **3a** localized at the Golgi.

Taken together, these results show that the fundamental machinery of Golgi targeting is preserved between the imaging and therapeutic versions of our molecules. We have revised the manuscript to clarify this point.

Original:

To investigate the morphological and functional consequences of such modulation, we used compound **3a**, a representative CyMA-d precursor that undergoes palmitoylation (Supplementary Fig. S4b) and accumulates at the Golgi (Supplementary Fig. S4c).

Supplementary Figure S4. CyMA-d accumulates at the Golgi and gets palmitoylated. (a) Cell viability of HeLa cells treated with **1a** for 24 hours. (b) HRMS of deacylated and palmitoylated CyMA (**3a**) in cell lysate. (c) CLSM of HeLa cells treated with an analogue of **3a** (500 nM), in which the benzene ring is replaced with a hydrophobic fluorophore for 8, 16, 32 minutes. Scale bar = 20 μm.

Revision:

To investigate the morphological and functional consequences of such modulation, we used compound **3a**, a representative CyMA-d precursor. To confirm that this non-fluorescent compound localizes to the Golgi, we provide two lines of evidence. First, a structurally similar fluorescent analogue (**3a'**), in which one benzene ring of the phenylalanine residue was replaced with an NBD fluorophore, showed clear and rapid accumulation at the Golgi (Supplementary Fig. S4b). Second, our LC-HRMS analysis of cell lysates confirmed that **3a** is palmitoylated (Supplementary Fig. S4c), providing strong biochemical evidence for its localization at the Golgi, where the required zDHHC enzymes reside⁷.

Supplementary Figure S4. CyMA-d accumulates at the Golgi and gets palmitoylated. (a) Cell viability of HeLa cells treated with **1a** for 24 hours. (b) CLSM of HeLa cells treated with **3a'** (500 nM), in which the benzene ring in **3a** is replaced with a hydrophobic fluorophore for 8, 16, 32 minutes. (c) HRMS of deacylated and palmitoylated CyMA (**3a**) in cell lysate. Scale bar = 20 μm.

[2- If the probe is trafficked to the Golgi independently of endocytosis, why is the uptake of the probe higher in the vps35 mutant, which has an endosome-to-Golgi trafficking defect, rather than being equal? More generally, I think it would be worthwhile looking at mutants of clathrin- and dynamin-related endocytosis.]

We thank the reviewer for the comments. The reviewer's question on the vps35 mutant result allows us to present a more detailed model that is consistent with this statement and our data.

We propose that the higher accumulation in the VPS35 mutant may result from altered trafficking dynamics rather than direct endosome-to-Golgi dependence. We revised the discussion on Vsp35 mutant and have briefly added this topic as an important direction for future studies in the revised Discussion section.

Original:

This result suggests that **1a** entry into the cell is energy dependent, but is unlikely to rely on endosome-to-Golgi transport, as evident by higher (rather than lower) cellular uptake in VPS35-mutant *Drosophila* hemocytes (Supplementary Fig. S1h).

Revision:

This result suggests that **1a** entry into the cell is energy-dependent but independent of the canonical Vps35-retromer pathway, as evident by higher (rather than lower) cellular uptake in VPS35-mutant *Drosophila* hemocytes (Supplementary Fig. S1h). This increased accumulation may result from the dual role of the Vps35 retromer complex, which helps traffic cargo from the endosome to both the Golgi and the plasma membrane for efflux. Since **1a** utilize a Vps35-independent mechanism to reach the Golgi, the blockage of the Vps35-dependent efflux pathway to the plasma membrane leads to a higher intracellular concentration of the probe, resulting in its increased accumulation at the Golgi.

Original:

Looking forward, the concept of establishing enzyme-controlled, futile assembly-disassembly cycles at defined subcellular locations and consuming endogenous metabolites provides a broadly applicable framework. Characterizing the precise nature of the observed subcellular reorganizations and their cargo will be a critical next step.

Revision:

Looking forward, the concept of establishing enzyme-controlled, futile assembly-disassembly cycles at defined subcellular locations and consuming endogenous metabolites provides a broadly applicable framework. To realize the full potential of this approach, several critical next steps are required. These include characterizing the precise nature of the observed subcellular reorganizations and their cargo, as well as undertaking a detailed quantitative analysis of this dynamic system. Furthermore, a detailed investigation into the various upstream endocytic pathways and specific proteins most affected by CyMA-d using genetic approaches would also be a valuable future direction.

[3-Many approaches rely on drug treatment, which I don't consider problematic. However, as suggested in my previous point on endocytosis, some conclusions may need to be supported by genetic approaches, as well as for ER-Golgi trafficking. Your data suggest that Golgi disruption occurs by mechanisms other than ER-Golgi redistribution, and indicate disruption to anterograde and retrograde transport. However, it would be good to have some genetic confirmation of this.]

We thank the reviewer for this thoughtful suggestion. We agree that genetic approaches are a powerful tool for confirming mechanisms and that these experiments would certainly add further value.

As suggested by the reviewer, for the key step of enzymatic activation, we employed a genetic approach: siRNA knockdown of the thioesterase enzymes (Figure 2e) to confirm their essential roles in the Golgi targeting. We agree that extending genetic confirmation to other trafficking pathways is an excellent direction. In fact, we have an ongoing project

dedicated to identifying the specific proteins most affected by CyMA, and these studies will include genetic validation. We now highlight this as an important future direction in the revised Discussion.

Original:

Looking forward, the concept of establishing enzyme-controlled, futile assembly-disassembly cycles at defined subcellular locations and consuming endogenous metabolites provides a broadly applicable framework. Characterizing the precise nature of the observed subcellular reorganizations and their cargo will be a critical next step.

Revision:

Looking forward, the concept of establishing enzyme-controlled, futile assembly-disassembly cycles at defined subcellular locations and consuming endogenous metabolites provides a broadly applicable framework. To realize the full potential of this approach, several critical next steps are required. These include characterizing the precise nature of the observed subcellular reorganizations and their cargo, as well as undertaking a detailed quantitative analysis of this dynamic system. Furthermore, a detailed investigation into the various upstream endocytic pathways and specific proteins most affected by CyMA-d using genetic approaches would also be a valuable future direction.

[4- Finally, my main point is on S-palmitoylation, which brings me back to my first point. N-palmitoylation is considered irreversible, whereas S-palmitoylation is reversible. There is therefore a discrepancy between this and the phenotypes observed with the NTG version of the probe. This probe, which disrupts the Golgi, has a strong effect on the cell, even leading to cell death. The authors demonstrate that palmitoylation is responsible for the probe's inability to diffuse at the Golgi. At the same time, I would expect a less drastic effect on the cell if palmitoylation were reversible.]

We thank the reviewer for this insightful comment, which highlights the central concept of our work, that is, how a chemically reversible process at the molecular level can generate potent and sustained cellular effects.

We believe the reversibility of S-palmitoylation plays a key role in driving the observed phenotype. Continuous palmitoylation/depalmitoylation creates a futile cycle that persistently consumes palmitoyl-CoA, placing a sustained metabolic burden on the cell and disrupting the lipidation of essential endogenous proteins. At the same time, this cycling dynamically maintains solid-like assemblies at the Golgi, as evidenced by our FRAP data showing a high immobile fraction. These nanostructures physically obstruct Golgi function, severely impairing trafficking and signaling.

We interpret these results to indicate that the reversibility of S-palmitoylation contributes to the observed phenotype by enabling metabolic drain and persistent Golgi obstruction

overwhelms homeostasis, ultimately leading to cell death. We have revised the Discussion to make this central point more clearly.

Original:

This unique mechanism of action provides a fundamentally different paradigm from conventional inhibitors and is what endows these intracellular materials with significant translational value. While traditional therapeutic strategies often rely on specific binding, they are vulnerable to resistance. In contrast, CyMA-d leverages a well-conserved enzymatic process for its activation.

Revision:

This unique mechanism of action provides a fundamentally different paradigm from conventional inhibitors and is what endows these intracellular materials with significant translational value. While traditional therapeutic strategies often rely on specific binding, they are vulnerable to resistance. In contrast, CyMA-d leverages a well-conserved enzymatic process for its activation. The reversibility of S-palmitoylation is likely what drives the strong phenotype. The continuous palmitoylation/depalmitoylation creates a futile cycle that persistently consumes palmitoyl-CoA, placing a sustained metabolic burden on the cell. At the same time, this cycling dynamically maintains the solid-like assemblies at the Golgi, which physically obstruct its function, severely impairing trafficking and signaling. Thus, the potent cellular effects arise not despite the reversibility, but because of it; the combination of metabolic drain and physical obstruction overwhelms cellular homeostasis, ultimately leading to cell death.

[Another point is that the authors observed a general reduction in palmitoylation with the NTG version of the probe, which is palmitoylated itself. Does the probe detach from the Golgi at some point? What is the lifetime of the NTG probe at the Golgi?]

We thank the reviewer for this insightful comment. CyMA is indeed both a substrate for palmitoylation and, over time, a disruptor of the process. In the early stages, **3a** engages the palmitoylation/depalmitoylation cycle to accumulate and assemble at the Golgi, while the global reduction in endogenous palmitoylation is a downstream effect of Golgi disruption. As a small molecule, CyMA is expected to undergo faster palmitoylation/depalmitoylation kinetics than proteins, enabling dynamic turnover. Accordingly, CyMA can transiently detach via depalmitoylation, and its effective lifetime at the Golgi reflects the balance between detachment and the now-impaired re-palmitoylation on disrupted membranes.

As shown in Supplementary Figure S12, prolonged treatment with **3a** leads to aberrant Golgi-derived structures that are likely to serve as the main sites of this turnover. The precise lifetime of the probe within these structures is the focus of ongoing work, and we have clarified this in the revised Discussion section.

Original:

The accumulation of CyMA-d assemblies physically disrupts essential Golgi functions. More than simple organelle disruption, this process culminates in a profound reorganization of Golgi-derived membranes into distinct, aberrant subcellular structures (Supplementary Fig. S12). The emergence of these *de novo* compartments appears to be a critical mechanistic event, likely driving cytotoxicity through complex interactions with intrinsic cellular machinery.

Revision:

The accumulation of CyMA-d assemblies physically disrupts essential Golgi functions. More than simple organelle disruption, this process culminates in a profound reorganization of Golgi-derived membranes into distinct, aberrant subcellular structures (Supplementary Fig. S12). Our results suggest that CyMA-d acts as both a substrate for palmitoylation and, over time, a disruptor of the process. In the early stages, 3a engages the palmitoylation/depalmitoylation cycle to accumulate at the Golgi, while the global reduction in endogenous palmitoylation is a downstream effect of this disruption. As a small molecule, CyMA is expected to undergo faster kinetics than proteins, enabling a dynamic turnover where it can transiently detach via depalmitoylation. Its effective lifetime at Golgi therefore reflects the balance between detachment and the now-impaired re-palmitoylation on these disrupted membranes. These aberrant Golgi-derived structures likely serve as the main sites of this turnover, and determining the precise lifetime of the probe within them will be a focus of future work. The sustained presence of these active, *de novo* compartments, which are dynamically maintained by this futile cycle, appears to be the critical mechanistic event that drives cytotoxicity through complex interactions with intrinsic cellular machinery.

[The data on RAS, AKT and mTOR at the end of the paper suggest that cell death and proliferation defects are due to a general decrease in palmitoylation, but are all these defects linked to the Golgi? DHHCs localize to different membranes within the cell: the ER (DHHC1 and 6), the ER-Golgi (DHHC2 and 9), the Golgi (DHHC7 and 8), and even the plasma membrane (DHHC5, 20, and 21). How can you therefore be completely sure that the effects are linked to the Golgi only?]

We thank the reviewer for the comments. We agree that it is unlikely that all these defects are linked to the Golgi. While we cannot formally exclude contributions from other sites, the convergence of these data provides strong support for Golgi as the main driver. To clarify this point, we added a few sentences in revision.

Addition:

While the non-fluorescent CyMA-d probe cannot be directly visualized and could in principle localize to other membranes where palmitoylation occurs, several evidences support that Golgi disruption results in the observed phenotypes. First, fluorescent version of CyMA-d confirmed direct targeting by selectively accumulating in the Golgi. Second, Golgi's role as the central hub for S-palmitoylation⁷ makes it the most favorable site for CyMA's activity. Third, the phenotypes follow a clear cascade where Golgi damage impairs trafficking and downstream signaling. Although contributions from other organelles cannot be ruled out, the available evidence is most consistent with Golgi disruption as the predominant effect.

[You cannot see the NTG probe, so it could localize to other membranes besides the Golgi. Palmitoylation occurs not only at the Golgi, and the phenotypic effects could be due to processes other than those related to the Golgi.]

We thank the reviewer for the comments. We agree that the NTG probe cannot be directly visualized, and in principle could localize to other membranes where palmitoylation occurs. We acknowledge this limitation, but note that complementary data from the fluorescent analogues suggest predominant Golgi localization. We have clarified this caveat and rationale in the revised Discussion.

Addition:

While the non-fluorescent CyMA-d probe cannot be directly visualized and could in principle localize to other membranes where palmitoylation occurs, several evidences support that Golgi disruption results in the observed phenotypes. First, fluorescent version of CyMA-d confirmed direct targeting by selectively accumulating in the Golgi. Second, Golgi's role as the central hub for S-palmitoylation⁷ makes it the most favorable site for CyMA's activity. Third, the phenotypes follow a clear cascade where Golgi damage impairs trafficking and downstream signaling. Although contributions from other organelles cannot be ruled out, the available evidence is most consistent with Golgi disruption as the predominant effect.

[Overall, I really liked this paper, which I think is supported by strong evidence and quantification. However, I would encourage the authors to clarify the final point that I raised, otherwise this probe will be used in future papers to disrupt the function of the Golgi apparatus and draw conclusions from this. However, this could alter much more than the Golgi apparatus, which could lead to inappropriate statements being made.]

We thank the reviewer for the thoughtful comments. We are very grateful to the reviewer for the positive assessment and strong support for our work.

We appreciate the reviewer's concern and have revised the discussion to more clearly delineate between CyMA-i (for imaging) and CyMA-d (for disruption).

Addition:

Our division of the platform into CyMA-i for imaging and CyMA-d for disruption is a first step toward ensuring these tools are used with precise intent. To further enhance this specificity, we are actively developing next-generation CyMA variants designed to be more sensitive and minimally disruptive, so that the imaging and perturbation functions can be better separated, which will further strengthen the utility and precision of the platform in future applications.

The following are our responses to the comments (in Italics) of the reviewer and the changes (underlined) in the manuscripts.

A) Reviewer 1:

[As I already mentioned in my first review, overall, this is a good work, with a substantial amount of interesting data. The reported results may be of interest to the scientific community, and therefore, it may deserve publication. However, some of the points I raised in my previous review have not been clarified yet.]

We thank the reviewer for their continued evaluation of our manuscript and for reiterating the value of our work. We have made further changes to address the remaining concerns.

[The authors have included the structural characterization for compounds 1a and 3a, but the ones corresponding to the other synthesized compounds (2b, 2c, 2d, 4a-4c, 5, 6, DAN-CyMA-i and DBD-CyMA) are still lacking. A correct characterization of synthesized molecules is always required, and this is even more important if their biological effect is studied.]

We thank the reviewer for the comments. We agree with the reviewer that the correct characterization of synthesized molecules is essential.

While we have provided NMR spectra for the key lead compounds (**1a** and **3a**) as previously noted, NMR analysis is technically unsuitable for characterizing the full panel of peptide analogues and controls (**2b-2d**, **4a-4c**, **5**, **6**, **DAN-CyMA-i**, and **DBD-CyMA-i**) because of the multiple conjugating systems of these peptide derivatives, which results in significant line broadening and complex spectra that preclude definitive structural assignment via NMR.

To ensure rigorous identification despite these limitations, we have carefully updated and verified our spectral data to meet the standard of Nature Communications requirements by using Liquid Chromatography-High Resolution Mass Spectrometry (LC-HRMS) in this revision. Specifically, we verified that all experimentally observed mass peaks < 0.003 Da error range of the calculated theoretical exact mass. In addition, we confirmed that the observed isotopic patterns match the theoretical patterns predicted for each compound's elemental formula.

The HRMS data, combined with the correct isotopic fingerprinting, provides sufficient and unambiguous evidence for the identity of the synthesized compounds. These data are now fully detailed in the revised Supplementary Figures S13 and S14 and the Methods section.

Original in SI:

Supplementary Figure S13. LC-HRMS of the synthesized CyMA-i, **1a**, **2a-2d**, **5**, DAN-CyMA-i, DBD-CyMA-i.

Revision in SI:

Supplementary Figure S13. LC-HRMS of the synthesized CyMA-i, **1a**, **2a-2d**, **5**, DAN-CyMA-i, DBD-CyMA-i, and **palm-1a**.

Original in SI:

Supplementary Figure S14. LC-HRMS of the synthesized CyMA-d, **3a**, **4a-4c**, **6**.

Revision in SI:

Supplementary Figure S14. LC-HRMS of the synthesized CyMA-d, **3a**, **4a-4c**, **6**.

Addition in SI:

Synthesized CyMA characterized by LC/HR-MS

The analysis was performed on a C18 reverse phase column using water (0.1% formic acid) and acetonitrile (0.1% formic acid) as the mobile phases, with a gradient of 1% to 99% acetonitrile over 14 minutes. The identity of each compound was confirmed by matching the experimentally observed high-resolution mass-to-charge ratio (m/z) with the calculated theoretical exact mass. We further verified these assignments by confirming that the observed isotopic distribution pattern of the peaks in the mass spectrum matched the theoretical pattern predicted for the compound's elemental formula.

[Moreover, for the identification of palmitoylated Peptide-SPalm resulting from different CyMA (Figure S1k), can you be sure that the palmitoylated peptides will be detectable under the used conditions (C18 column)? Do you have synthesized samples to analyse them and compare their retention times with the ones observed in the cell lysates? The unambiguous detection of the

formed palmitoylated peptides is crucial for the project, and it may be challenging in a sample derived from a cell lysate if the required reference samples are lacking.]

We thank the reviewer for the comments. We agree that referencing a synthesized standard is crucial for validating the LC-MS results obtained from complex cell lysates.

Following the suggestion of the reviewer, we used chemically synthesized the palmitoylated reference compound, **palm-1a**, as a standard. We analyzed **palm-1a** under the same LC-HRMS conditions (using a C18 column with a water/acetonitrile gradient) as the cell lysate samples. As shown in the revised Supplementary Figure S11, the retention time of the synthesized **palm-1a** (bottom trace) matches that of the palmitoylated species detected in the cell lysate (top trace).

This retention time alignment, combined with the matching HRMS and isotopic patterns, confirms that the palmitoylated peptides are indeed detectable and identifiable under our experimental conditions. We have also updated the Methods section to include the synthetic procedure for **palm-1a**.

Original in SI:

Supplementary Figure S1. Non-diffusive CyMA specifically accumulate at the Golgi. (a-c) Colocalization study of CyMA with lysosome (a), mitochondria (b), ER (c). (d) CLSM of HeLa cells treated with **1a** (200 nM, 50 nM; 4, 8, 16 min.). (e)

SDCM of ManII-RFP-expressing *Drosophila* hemocytes (He-Gal4 driving UAS-ManII-RFP) treated with **1a** (5 μ M, 10 min.). (f) Molecular structure of MGCTLSA-NBD as a control, and CLSM of HeLa cells treated with MGCTLSA-NBD (10 μ M; 4, 8, 16 min.). (g) CLSM of HeLa cells treated with **1a** (500 nM; 1 hour) at different temperatures. (h) SDCM of WT *Drosophila* hemocytes or VPS35-mutant (*vps35e42/Df(Exel)6078*) *Drosophila* hemocytes treated with **1a** (5 μ M, 10 min.). (i) CLSM of HeLa cells pretreated with or without ML211 (50 μ M; 30 min.) or DC661 (20 μ M; 30 min.) or 2-BP (20 μ M; 30 min.) and then treated with C6-NBD-Ceramide (20 μ M; 16 min.). (j) Immunoblotting of PPT1, LYPLA1, LYPLA2 and GAPDH in HeLa cells transfected with the corresponding siRNA. (k) Representative extracted-ion chromatograms from the LC-HRMS analysis of HeLa cell lysates, showing the retention time comparison of the corresponding Peptide-SH and Peptide-SPalm resulting from different CyMA (**1a**, **2a**, and **3a**) using LC-HRMS. (l) LC trace of two starting materials, **2a** and 2-BP, and their mixture incubated for 4 h at 37 °C in PBS. (m) CLSM of HeLa cells pretreated with or without 2-BP (20 μ M; 30 min.) and then treated with **2a** (2 μ M; 30 min.). (n) Quantification of the fluorescence intensity at Golgi of zDHHC siRNA transfected HeLa cells treated with **1a** (500 nM) within 16 minutes. Scale bar = 20 μ m except for (g), where scale bar = 10 μ m.

Revision in SI:

Supplementary Figure S1. Non-diffusive CyMA specifically accumulate at the Golgi. (a-c) Colocalization study of CyMA with lysosome (a), mitochondria (b), ER (c). (d) CLSM of HeLa cells treated with 1a (200 nM, 50 nM; 4, 8, 16 min.). (e)

SDCM of ManII-RFP-expressing *Drosophila* hemocytes (He-Gal4 driving UAS-ManII-RFP) treated with **1a** (5 μ M, 10 min.). (f) Molecular structure of MGCTLSA-NBD as a control, and CLSM of HeLa cells treated with MGCTLSA-NBD (10 μ M; 4, 8, 16 min.). (g) CLSM of HeLa cells treated with **1a** (500 nM; 1 hour) at different temperatures. (h) SDCM of WT *Drosophila* hemocytes or VPS35-mutant (*vps35^{e42}/Df(Exel)6078*) *Drosophila* hemocytes treated with **1a** (5 μ M, 10 min.). (i) CLSM of HeLa cells pretreated with or without ML211 (50 μ M; 30 min.) or DC661 (20 μ M; 30 min.) or 2-BP (20 μ M; 30 min.) and then treated with C6-NBD-Ceramide (20 μ M; 16 min.). (j) Immunoblotting of PPT1, LYPLA1, LYPLA2 and GAPDH in HeLa cells transfected with the corresponding siRNA. (k) Representative extracted-ion chromatograms (EIC) from the LC-HRMS analysis of HeLa cell lysates, showing the retention time comparison of the corresponding Peptide-SH and Peptide-SPalm resulting from different CyMA (**1a**, **2a**, and **3a**) using LC-HRMS. (l) EIC of palmitoylated CyMA (**1a**) detected in the cell lysate and the LC of the chemically-synthesized **palm-1a**, which share similar retention time. (m) LC trace of two starting materials, **2a** and 2-BP, and their mixture incubated for 4 h at 37 °C in PBS. (n) CLSM of HeLa cells pretreated with or without 2-BP (20 μ M; 30 min.) and then treated with **2a** (2 μ M; 30 min.). (o) Quantification of the fluorescence intensity at Golgi of zDHHC siRNA transfected HeLa cells treated with **1a** (500 nM) within 16 minutes. Scale bar = 20 μ m except for (g), where scale bar = 10 μ m.

Original:

Liquid chromatography coupled to high-resolution mass spectrometry (LC-HRMS) confirmed CyMA lipidation at the Golgi by identifying peaks for both the deacylated product (Peptide-SH) and the palmitoylated CyMA (Peptide-SPalm) of **1a** (**Fig. 2f**, **Supplementary Fig. S1k**).

Revision:

Liquid chromatography coupled to high-resolution mass spectrometry (LC-HRMS) confirmed CyMA lipidation at the Golgi by identifying peaks for both the deacylated product (Peptide-SH) and the palmitoylated CyMA (Peptide-SPalm) of **1a** (**Fig. 2f**, **Supplementary Fig. S1k**). Notably, the palmitoylated CyMA exhibited a retention time comparable to that of the chemically synthesized **palm-1a** (**Supplementary Fig. S1l**), further confirming that CyMA undergo palmitoylation.

Addition in SI:

Synthesis of palm-1a

The synthesized 2a (12.1 mg) was dissolved in anhydrous TFA and chilled in ice-water bath. Then, 3 equivalent palmitoyl-chloride (16.2 mg) was added into the solution, and the reaction mixture was stirred at room temperature overnight. After TFA was removed by rotary evaporation, the resulting oily product was purified by silica gel flash chromatography (Hexane/Acetone, gradient from 1:0 to 2:1) to afford the desired product as a yellow solid.

[Regarding the discussion about the inability of the probe to diffuse at the Golgi and the dynamic palmitoylation cycle, pointed out in my first revision and also commented on by Reviewer 3, I am not convinced by the explanation provided.

A dynamic reversible S-palmitoylation associated with a high immobile fraction would require that the thioesterases are also located in the Golgi, which does seem to be the case (the main effect is observed with PPT1, which is mainly a lysosomal enzyme). All in all, palmitoylation and Golgi localization may be clear, but not its reversibility. This point is unclear, and the discussion sounds too speculative at the moment, and therefore it should be revised.]

We thank the reviewer for the comments. Following the reviewer's suggestion, we have revised the discussion and incorporated the following supporting evidence.

1. **PPT1 Localization.** Although PPT1 is predominantly lysosomal, multiple independent datasets demonstrate that it is not restricted to this compartment. The "Subcellular map of the human proteome" (10.1126/science.aal332) explicitly identifies PPT1 localization at the Golgi apparatus in addition to lysosomes. Furthermore, Segal-Salto et al. (10.1371/journal.pone.0146466) demonstrate that PPT1 is secreted and can be found in non-lysosomal compartments.
2. **PPT1 pH Profile.** The reviewer correctly notes that PPT1 functions in acidic environments. However, it retains catalytic activity at pH levels > 4.5, which is compatible with the Golgi environment (typically pH 6.0–6.7). Studies on NCL screening (10.1016/j.ymgme.2018.03.007) and neuronal health (10.3389/fnsyn.2019.00025) confirm PPT1 activity extends beyond the strict acidity of the lysosome.
3. **Inter-organelle Lipid Exchange Enables Cycling Regardless of PPT1's Primary Location.** Even under the most conservative assumption that PPT1 were strictly lysosomal, the reversible nature of CyMA would remain plausible. CyMA are small molecule lipid-mimics. There is extensive vesicular and non-vesicular lipid exchange between the Golgi and lysosomes (10.1016/j.tcb.2017.07.006). Through this natural trafficking, CyMA could undergo depalmitoylation in lysosomes and repalmitoylation upon returning to the Golgi. Therefore, reversibility does not strictly depend on fully Golgi-localized thioesterases.
4. **Confirming the Cycle (BODIPY-CyMA Chase).** To further validate the "futile cycle", we performed a pulse-chase experiment using two BODIPY-CyMA variants with distinct excitation/emission profiles but identical targeting mechanisms (detailed in our preprint bioRxiv "Cycling Molecular Assemblies

(CyMA) for Ultrasensitive Golgi Imaging", 10.1101/2025.11.19.689330, Figure 2E). Both variants share identical targeting and palmitoylation mechanisms. The experiment demonstrates that newly introduced probes displace pre-existing Golgi-bound CyMA by competing for zDHHC enzymes. This directly confirms that individual molecules cycle through palmitoylation–depalmitoylation events even though the higher-order assembly remains stable.

Addition:

PPT1's documented localization at the Golgi^{92,93} and its ability to retain enzymatic activity at pH > 4.5^{94,95} support a model in which CyMA undergoes local, *in situ* deacylation, enabling dynamic cycling at the Golgi. This localized enzymatic turnover drives a continuous assembly–disassembly loop and likely operates in parallel with cytosolic thioesterases such as LYPLA1/2, whose individual contributions we were unable to exclude due to inhibitor cytotoxicity. Importantly, these depalmitoylation pathways are not mutually exclusive. Even CyMA molecules that are depalmitoylated by PPT1 in lysosomes can return to the Golgi through extensive vesicular and non-vesicular lipid exchange⁹⁶, where they can be repalmitoylated. Likewise, other depalmitoylases, such as members of the ABHD family⁹⁷, may also contribute to CyMA cycling. Together, these routes reinforce a robust palmitoylation–depalmitoylation cycling mechanism. Furthermore, our chase experiments using a newly introduced Golgi probe⁹⁸ provide additional support for this dynamic cycling model.

Addition:

92. Thul, P. J. et al. A subcellular map of the human proteome. *Science* 356, eaal3321 (2017).

93. Segal-Salto, M., Sapir, T. & Reiner, O. Reversible Cysteine Acylation Regulates the Activity of Human Palmitoyl-Protein Thioesterase 1 (PPT1). *PLOS ONE* 11, e0146466 (2016).

94. Itagaki, R. et al. Characteristics of PPT1 and TPP1 enzymes in neuronal ceroid lipofuscinosis (NCL) 1 and 2 by dried blood spots (DBS) and leukocytes and their application to newborn screening. *Mol Genet Metab* 124, 64-70 (2018).

95. Koster, K. P. & Yoshii, A. Depalmitoylation by Palmitoyl-Protein Thioesterase 1 in Neuronal Health and Degeneration. *Frontiers in Synaptic Neuroscience* Volume 11 - 2019 (2019).

96. Thelen, A. M. & Zoncu, R. Emerging Roles for the Lysosome in Lipid Metabolism. *Trends Cell Biol* 27, 833-850 (2017).

97. Lin, D. T. S. & Conibear, E. ABHD17 proteins are novel protein depalmitoylases that regulate N-Ras palmitate turnover and subcellular localization. *eLife* 4, e11306 (2015).

98. Tan, W. et al. Cycling Molecular Assemblies (CyMA) for Ultrasensitive Golgi Imaging. *bioRxiv*, 2025.2011.2019.689330 (2025).

B) Reviewer 2:

[The authors answered all the concerns raised by the reviewer. This manuscript can be now considered for publication in this current form.]

We sincerely thank the reviewer for the constructive feedback throughout the process, which helped us improve this work.

C) Reviewer 3:

[The authors have now addressed my concerns to my satisfaction, either through experimental approaches, by revising the text, or by exercising more caution in their conclusions and clarifying the limitations of their work. I therefore have no further concerns.]

We sincerely thank the reviewer for the constructive feedback throughout the process, which helped us improve this work.

A) Reviewer 1.

[Reviewer 1: "The authors have addressed my concerns satisfactorily, and the manuscript has been revised accordingly. I recommend the acceptance of the manuscript."]

We thank the reviewer for the constructive feedback and agree the concerns of the reviewer have been addressed.